# t-SMILES: a fragment-based molecular representation framework for de novo ligand design

Juan-Ni Wu [1], Tong Wang[1], Yue Chen [1], Li-Juan Tang[1], Hai-Long Wu [1] ✉ & Ru-Qin Yu [1] ✉

Effective representation of molecules is a crucial factor affecting the performance of artificial intelligence models. This study introduces a flexible, fragment-based, multiscale molecular representation framework called t-SMILES (tree-based SMILES) with three code algorithms: TSSA (t-SMILES with shared atom), TSDY (t-SMILES with dummy atom but without ID) and TSID (t-SMILES with ID and dummy atom). It describes molecules using SMILES-type strings obtained by performing a breadth-first search on a full binary tree formed from a fragmented molecular graph. Systematic evaluations using JTVAE, BRICS, MMPA, and Scaffold show the feasibility of constructing a multi-code molecular description system, where various descriptions complement each other, enhancing the overall performance. In addition, it can avoid overfitting and achieve higher novelty scores while maintaining reasonable similarity on labeled low-resource datasets, regardless of whether the model is original, data-augmented, or pre-trained then fine-tuned. Furthermore, it significantly outperforms classical SMILES, DeepSMILES, SELFIES and baseline models in goal-directed tasks. And it surpasses state-of-the-art fragment, graph and SMILES based approaches on ChEMBL, Zinc, and QM9.

Many molecular descriptors have been used in molecular modeling tasks. However, unlike natural language processing (NLP) and image recognition, where deep learning has shown exceptional performance, one of the domain-specific challenges for artificial intelligence (AI) assisted molecular discovery is the lack of a naturally applicable, complete and "raw" molecular representation[1]. As molecular representation determines the content, nature and interpretability of the chemical information retained (e.g., physicochemical properties, pharmacophores, functional groups), how the molecule is represented becomes a limiting factor in the performance and explicability of AI models[1].

In recent years, various AI models have been proposed to handle molecular discover tasks, such as automatically generating molecules based on different molecular descriptors. Among them, models with sequence representations[2–4], such as Simplified Molecular Input Line Entry System (SMILES)[5], and graphs[6] are the most popular. At the same time, a plethora of models generating molecules in 3D[7] also starts to attract attention.

As a relatively more natural representation of molecules, a graph neural network (GNN) model could generate 100% valid molecules as it can easily implement valence bond constraints and other verification rules. However, it has been shown that the expressive power of standard GNNs is bounded by Weisfeiler-Leman graph isomorphism phenomenon, the lack of ways to model long-range interactions and higher-order structures limited the use of GNNs[8], though some recent studies have proposed new methods such as subgraph isomorphism[9], message-passing simple networks[10] and many other techniques to improve the expressive power of standard GNNs[11].

SMILES is a linear string obtained by performing a depth-first search on a molecular graph. When generating SMILES, parentheses

[1]State Key Laboratory of Chemo/Biosensing and Chemometrics, College of Chemistry and Chemical Engineering, Hunan University, Changsha 410082, PR China. ✉e-mail: hlwu@hnu.edu.cn; rqyu@hnu.edu.cn

and two numbers must occur in pairs with deep nesting to represent molecular topological structure, such as branches and rings. Models trained on SMILES generate some chemically invalid strings, especially when trained on small datasets. Although it is a straightforward task to select valid strings for output, the critical concern is that these invalid strings indicate that even state-of-the-art (SOTA) deep learning models struggle with accurately comprehending SMILES syntax and semantics, one of the main reasons being unbalanced parentheses and ring identifiers. This is considered as a limitation that needs to be addressed[12].

Two alternative solutions to the classical SMILES have been proposed later. DeepSMILES (DSMILES)[13] resolves most cases of syntactical mistakes caused by long-term dependencies. However, it still allows for semantically incorrect strings, such as "CO=CC" (the oxygen atom that has three bonds−a violation of the maximum number of bonds that neutral oxygen can form), and some studies indicate that the advanced grammar has a detrimental effect on the learning capability in some specific tasks[14,15]. Self-referencing embedded strings (SELFIES)[16] is an entirely different molecular representation, in which every SELFIES string specifies a valid chemical graph. However, the approach's focus on robustness can make certain SELFIES more challenging to read[15].

SMILES, DSMILES, and SELFIES are all atom-based linear representations. Compared with atom-based techniques, the search space is significantly reduced by using the fragment strategy. In addition, fragments could provide fundamental insights into molecular recognition, for example, between proteins and ligands. Consequently, there is a higher probability of finding molecules that match the known targets. Currently, almost all fragment (motif or substructure) based deep learning[17] methods published rely on a specific substructure dictionary of candidate fragments[18−25]. It is obvious that dictionary-ID-based models suffer from some fundamental problems such as in-vocabulary, out-of-vocabulary, and high-dimensional sparse representation (curse of dimensionality). Although deep learning has been widely used in molecular generation tasks[26,27] with the fragment-based method, the approach of fragmenting molecules and encoding molecular substructures as a string-type sequence to finally generate new molecules has not yet been thoroughly explored.

Jean-Marie Lehn's famous analogy "Atoms are letters, molecules are the words, supramolecular entities are the sentences and the chapters"[28] was cited by the researchers[29] studying the rank distribution of fragments in organic molecules being similar to that of words in the English language. Furthermore, some investigations suggest that language model (LM) may outperform most GNNs in learning large and complex molecules[30]. And recently, Transformers[31] based LMs have demonstrated their ability to generate text that closely resembles human writing. These ideas inspired us to select the classical SMILES as the starting choice for fragment description and adopt advanced NLP techniques to handle fragment-based molecular modeling tasks, which could hybridize the advantages of graph model paying more attention to molecular topology structure and LM having powerful learning ability.

Therefore, we propose a new molecule description framework called t-SMILES based on fragmented molecule, which describes a molecule with a SMILES-type string and can take the sequence-based models as the primary generation model. This study introduces three t-SMILES code algorithms: TSSA (t-SMILES with shared atom), TSDY (t-SMILES with dummy atom but without ID) and TSID (t-SMILES with ID and dummy atom).

The newly proposed t-SMILES framework first generates an acyclic molecular tree (AMT) whose role is to represent fragmented molecules, as illustrated in Fig. 1. The AMT is transformed into a full binary tree (FBT) in the second stage. Finally, the breadth-first traversal of the FBT yields a t-SMILES string.

Compared to SMILES, t-SMILES introduces only two new symbols, "&" and "^", to encode multi-scale and hierarchical molecular topologies. So, the t-SMILES algorithm provides a scalable and adaptable framework that theoretically capable of supporting a broad range of substructure schemes, as long as they generate chemically valid fragments and produce valid AMT. Moreover, owing to its multiscale and hierarchical representation, the model based on t-SMILES is capable of learning high-level topology structural information while processing detailed substructure information. Notably, the t-SMILES algorithm can construct a multi-code system for molecule description. In this system, classical SMILES can be integrated as a special case of t-SMILES termed TS_Vanilla, and multiple descriptions can collaborate to improve comprehensive performance.

Well-trained string-based LMs have been proven to be effective in various molecular studies. The differences in performance among various codes depend deeply on their underlying distinction. Consequently, we methodically assess t-SMILES by initially delving into its distinctive features. Subsequently, we conduct experiments on two labeled low-resource datasets, JNK3[32] and AID1706[33] using TSSA and TSDY. Our investigation focuses on t-SMILES and its substitutes' limits, achieved via utilization of standard, data augmentation, and pre-training fine-tuning models. In line with our goal, we evaluate twenty goal-directed tasks on ChEMBL using TSDY, TSSA and TSID in parallel. We also thoroughly experiment on ChEMBL, Zinc, and QM9, via comparison between t-SMILES and their alternatives using similar settings. In addition, we compare various fragment-based baseline models and the SOTA GNN models. Lastly, an ablation study is carried out to confirm the effectiveness of the generative model based on SMILES with reconstruction. In order to evaluate the adaptability and flexibility of t-SMILES algorithm, four previously published fragmentation algorithms were utilized to break down molecules, including JTVAE[34], BRICS[35], MMPA[36] and Scaffold[37]. Three types of metrics: distribution-learning benchmarks, goal-directed benchmarks and Wasserstein distance metrics for physicochemical properties are used in different experiments.

Detailed comparative experiments demonstrate that the t-SMILES models have the potential to achieve 100% theoretical validity and generate highly novel molecules, outperforming SOTA SMILES-based models. Compared to SMILES, DSMILES, and SELFIES, the overall solution of t-SMILES can avoid the overfitting problem and significantly enhances balanced performance on low-resource datasets, regardless of whether data augmentation or pre-trained then fine-tuned models are used. In addition, t-SMILES models are proficient in capturing the physicochemical properties of molecules, ensuring that the generated molecules maintain similarity to the training molecule distribution. This results in significantly improved performance compared to existing fragment-based and graph-based baseline models. In particular, t-SMILES models with goal-directed reconstruction algorithm show considerable benefits over SMILES, DSMILES, SELFIES, and SOTA CReM[38] in goal-oriented tasks.

## Results

Various singleton and hybrid strategies can be used to achieve different goals in the t-SMILES family. Due to the limited computational resources, we evaluated TSSA and TSDY individually in two goal-oriented tasks after discussing their distinctive features. Furthermore, we conduct rigorous and systematic comparative experiments to assess singleton and hybrid TSSA, TSDY, and TSID models on ChEMBL, ZINC, and QM9.

Supplementary (SI) Table 6 presents a preliminary summary of t-SMILES coding algorithms and experiments. In SI.E.8, we report a brief study on ring-opening problem using TSID. SI.D.8 outlines the results of an ablation study. For more detailed information on experimental configurations and evaluation metrics, please refer to section SI.B.

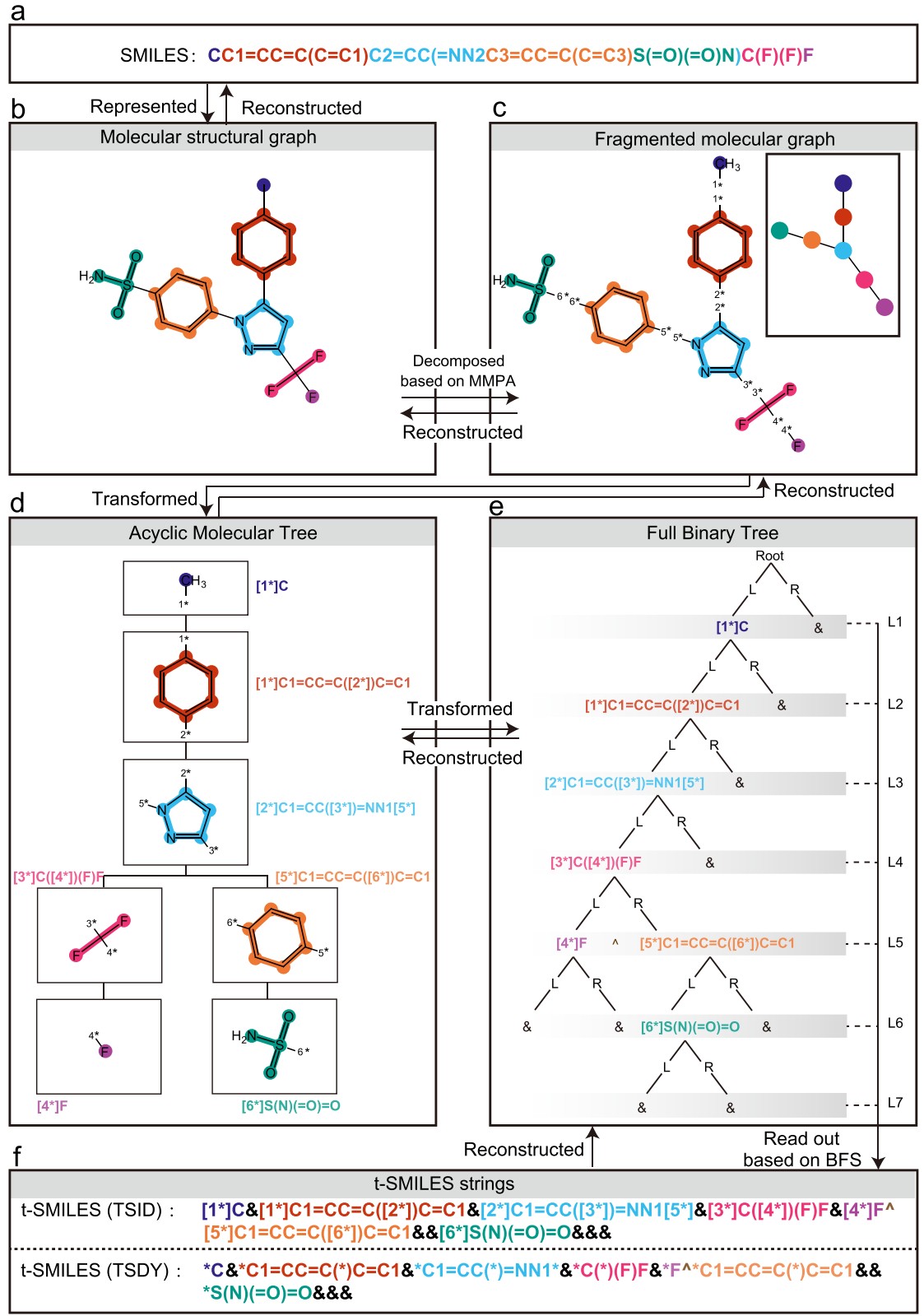

**a** SMILES: CC1=CC=C(C=C1)C2=CC(=NN2C3=CC=C(C=C3)S(=O)(=O)N)C(F)(F)F

**b** Molecular structural graph — Represented / Reconstructed

**c** Fragmented molecular graph — Decomposed based on MMPA / Reconstructed

**d** Acyclic Molecular Tree — Transformed / Reconstructed

[1*]C

[1*]C1=CC=C([2*])C=C1

[2*]C1=CC([3*])=NN1[5*]

[3*]C([4*])(F)F          [5*]C1=CC=C([6*])C=C1

[4*]F          [6*]S(N)(=O)=O

**e** Full Binary Tree — Transformed / Reconstructed — Read out based on BFS

Root — L R

L1  [1*]C  &
L2  [1*]C1=CC=C([2*])C=C1  &
L3  [2*]C1=CC([3*])=NN1[5*]  &
L4  [3*]C([4*])(F)F  &
L5  [4*]F  ^  [5*]C1=CC=C([6*])C=C1
L6  &  &  [6*]S(N)(=O)=O  &
L7  &  &

**f** t-SMILES strings — Reconstructed

t-SMILES (TSID) : [1*]C&[1*]C1=CC=C([2*])C=C1&[2*]C1=CC([3*])=NN1[5*]&[3*]C([4*])(F)F&[4*]F^[5*]C1=CC=C([6*])C=C1&&[6*]S(N)(=O)=O&&&

t-SMILES (TSDY) : *C&*C1=CC=C(*)C=C1&*C1=CC(*)=NN1*&*C(*)(F)F&*F^*C1=CC=C(*)C=C1&&*S(N)(=O)=O&&&

## Distinctive properties of t-SMILES

The training data includes essential information for designing efficient deep-learning algorithms. This section presents a brief overview of the token distribution and nesting depth for SMILES, DSMILES, SELFIES, and t-SMILES, serving as a valuable reference and a starting point for future research.

## Token distribution

LM uses probability and statistical techniques to calculate the possibility of word sequences in sentences and then do estimation[39]. Although t-SMILES introduces just two extra characters, "&" and "^", its token distribution differs entirely from SMILES. The distribution of tokens on zinc is shown in Fig. 2, more detailed information in SI.C.1.

**Fig. 1 | Overview of the t-SMILES algorithm. a** SMILES of molecule Celecoxib. **b** The structural formula of Celecoxib which is marked with different colors to indicate its fragments. **c** The molecule is broken down into fragments, each marked with a different color. The thumbnail illustrates the overall topological structure of fragmented molecule. **d** AMT(Acyclic Molecular Tree) of fragmented molecule. Each fragment is presented using both its SMILES code and structural formula. **e** FBT (Full Binary Tree) of fragmented molecule. Tree node is presented with fragment or new introduced symbl "&". "L" and "R" refer to the left or right sub-tree. "L1"–"L7" refers to the number of layers of FBT. In L5, new symbol '^' is used to separate two pieces in t-SMILES string. **f** TSID (t-SMILES with ID and dummy atom) and TSDY (t-SMILES with dummy atom but without ID) code of Celecoxib. The

colors in the t-SMILES string are used to match the corresponding fragments in the structural formula.The molecular graph is first decomposed using selected molecular fragmentation algorithm to build an AMT, which is then transformed into an FBT. Finally, the BFS (Breadth-First Search) algorithm is used to traverse the FBT and obtain its t-SMILES string. To reconstruct the molecule, rebuilding the FBT from the t-SMILES string, transforming it to an AMT, and finally assembling the AMT back into the original molecule. In TSID, [n*] are used to indicate joint point. When the IDs are removed from the TSID code, the TSDY code is created. New symbol "&" is used to mark empty tree nodes. TSSA (t-SMILES with shared atom) uses a different way to get pieces, please see Supplementary A.1 for the entire process and more examples. MMPA is used as an example in both figures to cut molecules.

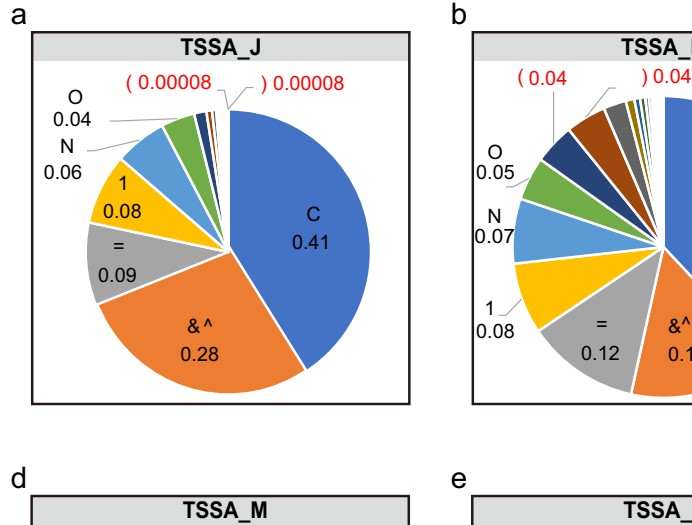

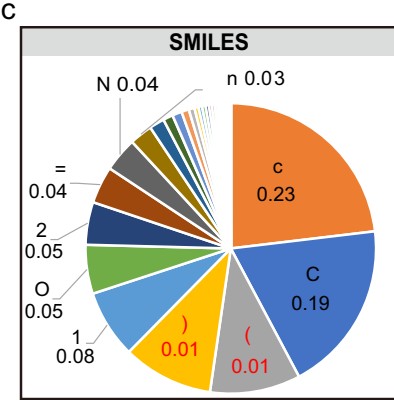

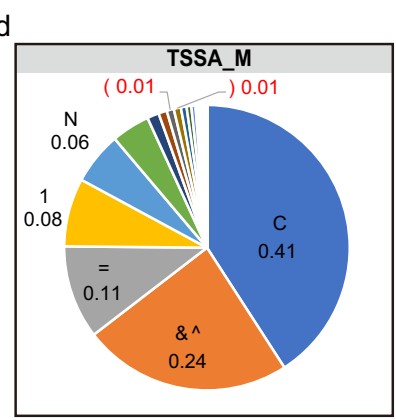

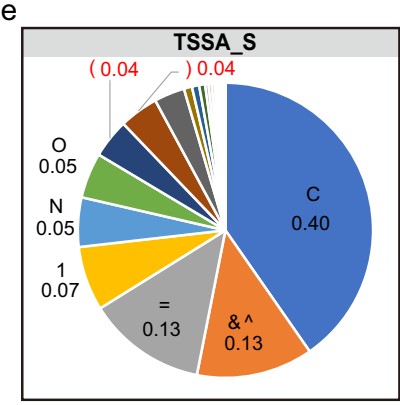

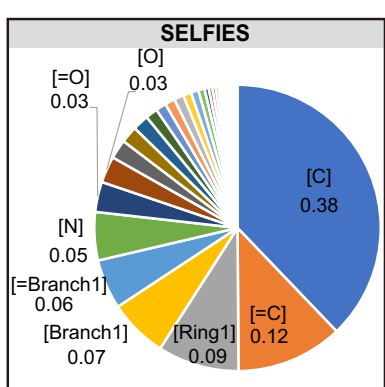

**Fig. 2 | Distributions of tokens for TSSA codes, SMILES, and SELFIES.** **a**, **b**, **c**, **d**, **e** and **f** are the token distributions of TSSA_J, TSSA_B, SMILES, TSSA_M, TSSA_S and SELFIES, respectively. The symbols "&^", which are used to indicate the molecular topology structure in t-SMILES, exhibit the second-highest frequency in the TSSA codes. However, they are not required to be in pairs, unlike the "(" and ")"

symbols in SMILES, which must be paired. The number of paired parentheses (highlighted in red) in t-SMILES codes exhibited a notable decline as they are limited to sub-fragments rather than the entirety of the SMILES string. The suffix letters "J, B, M, S" in various t-SMILES code names represent the fragmentation algorithm: JTVAE, BRICS, MMPA, and Scaffold.

Figure 2 shows that the proportions of the characters "(" and ")" in t-SMILES codes are all less than 5% (TSSA_J: 0.008%, TSSA_B: 4.3%, TSSA_M: 0.8%, TSSA_S: 4.2%. The suffix letters "J", "B", "M" and "S" in various TSSA, TSDY and TSID code names represent the fragmentation algorithm: JTVAE, BRICS, MMPA, and Scaffold.). In comparison, the proportions of characters "(" and ")" are more than 10% in SMILES-based code. On the other hand, the distribution shows that the symbols "&^", which are used to indicate the molecular topology structure in t-SMILES but do not have to be in pairs, have the second-highest frequency in the TSSA codes, just below the frequency of "C".

The comparison between t-SMILES and SMILES indicate that, a crucial task of the SMILES model is to learn and predict PAIRED "(" and ")" symbols. Conversely, the t-SMILES model must learn how to reason NON-PAIRED symbols "&" and "^". It is worth noting that, the addition

of symbols "&^" does not present a new challenging issue, like the scarcity reasoning problem, as evidenced by their high frequency.

### Nesting depth
How deep learning models such as LSTM[40] or Transformer[31] learn long-term dependencies is one of the driving forces behind the development of NLP. In classical SMILES, deeply nested pairs of characters, such as "(" and ")", are one of the main reasons why the generated string could not be decoded into a chemically valid molecule. Although SMILES is still used to represent molecular substructure, t-SMILES cuts molecules into small pieces and abandons the algorithm of depth-first traversal of molecular trees, which fundamentally reduces the depth of nesting and the proportion of characters that must appear in pairs.

For example, when compared to SMILES, the TSDY_M (MMPA-based TSDY) code on ChEMBL increases the proportion of the 0-1-2 nesting depth from 68.006% to 99.270%, while decreasing the 3-4-5 nesting depth from 31.886% to 0.730%, and the 6–11 nesting depth from 0.108% to 0.00019%. The detailed statistical data on the nesting depth of ChEMBL and Zinc are listed in SI.C.2.1 and SI.C.2.2, respectively.

## Reconstruction and generative model without training

MOG[41] argued that the common "pitfall" of existing molecular generative models based on distributional learning is that exploration is limited to the training distribution, and the generated molecules exhibit "striking similarity" to molecules in the training set. The authors point out: "models that do not require training molecules are free from this problem, but they introduce other problems such as long training time, sensitivity to the balance between exploration and exploitation, large variance, and most importantly, lack of information about the known distribution". From this point of view, t-SMILES provides a comprehensive solution by integrating distributional and non-distributional learning into a single system, thereby circumventing the aforementioned "pitfall".

The overall performance of t-SMILES model is determined by two primary factors: 1) the model's ability to efficiently learn and generate t-SMILES strings, and 2) the effectiveness of the reconstruction and molecular optimization algorithm. In t-SMILES, novel molecules can be generated by direct reconstruction of FBT without training. If we output all possible assembly results, we could generate a group of molecules from a set consisting of several molecular fragments with different structures. From this point of view, generating new molecules with the desired structure (desired properties) rather than duplicating the training set is the potential goal of the molecule generation task and not a negative aspect. The results of the random reconstruction on ChEMBL, Zinc, and QM9 are shown in Table 1 and SI.E.1-SI.E.3.

Tables and figures show that the direct reconstruction algorithm effectively preserves the physiochemical properties of the original dataset, as demonstrated by significantly high KLD (Kullback-Leibler divergence) and FCD (Fréchet ChemNet Distance) values across all three datasets. In addition, the uniqueness and novelty scores for TSSA on ChEMBL and Zinc exceed 0.5 for all four decomposition algorithms, suggesting that the reconstruction algorithm can serve as an efficient generation model without training.

Although novelty scores for TSDY have intentionally been reduced, it is still possible for TSDY to function as a generation model without training in certain conditions. However, it is needed for TSID to undergo training in order to build a deep neural network-based generative model. Despite the theoretical TSID reconstruction novelty score being zero, our attempts to represent molecules in the Kekule style resulted in scores of 0.004 and 0.010 on ChEMBL and Zinc. It is challenging to convert particular fragments and molecules into Kekule style using RDKit. This issue does not pose a problem for generative models, but we are actively working to enhance it for retrosynthesis and reaction prediction.

## Chemical space exploration and data augmentation for t-SMILES

One of the fundamental motivation for using fragments is their ability to sample the chemical space efficiently[42]. In t-SMILES system, training data can be augmented from four levels: 1) decomposition algorithm, 2) reconstruction, 3) enumeration of fragment strings and 4) enumeration of FBTs.

Different molecular segmentation algorithms divide a molecule in various ways, resulting in distinct segment groups. Subsequently, these segment groups cover different chemical spaces when producing new molecules. Refer to SI.C.3 and SI.G for further information on physicochemical properties and atom environment substructure. SI.C.4 displays the chemical spaces of different fragmentation algorithms in TSSA, which are directly reconstructed from FBTs for individual molecules.

In terms of data augmentation, recent evidence suggests that SMILES enumeration may be a valid way to augment the training data in a "small" dataset in order to improve the quality of generative models[14]. However, ref. 12 argues that "over-enumeration" in large

**Table 1 | Results by directly reconstructing molecules using random reconstruction algorithm**

| Code Algorithm | Dataset | Model | Valid | Unique | Novelty | KLD | FCD | FBTs |
|---|---|---|---|---|---|---|---|---|
| TSSA | ChEMBL | TSSA_J | 1.000 | 0.982 | 0.833 | 0.974 | 0.704 | 5029 |
| | | TSSA_B | 1.000 | 0.992 | 0.682 | 0.981 | 0.720 | 610633 |
| | | TSSA_M | 1.000 | 0.993 | 0.856 | 0.986 | 0.823 | 88938 |
| | | TSSA_S | 1.000 | 1.000 | 0.882 | 0.969 | 0.816 | 515329 |
| | Zinc | TSSA_J | 1.000 | 0.971 | 0.835 | 0.985 | 0.827 | 61786 |
| | | TSSA_B | 1.000 | 0.971 | 0.755 | 0.975 | 0.740 | 1197 |
| | | TSSA_M | 1.000 | 0.970 | 0.842 | 0.988 | 0.858 | 18989 |
| | | TSSA_S | 1.000 | 0.976 | 0.876 | 0.972 | 0.840 | 485 |
| | QM9 | TSSA_J | 1.000 | 0.929 | 0.304 | 0.977 | 0.971 | 279 |
| | | TSSA_B | 1.000 | 0.945 | 0.056 | 0.998 | 0.983 | 32 |
| | | TSSA_M | 1.000 | 0.911 | 0.141 | 0.997 | 0.979 | 238 |
| | | TSSA_S | 1.000 | 0.898 | 0.156 | 0.996 | 0.975 | 69 |
| TSDY | ChEMBL | TSDY_B | 1.000 | 0.996 | 0.210 | 0.997 | 0.915 | 4267 |
| | | TSDY_M | 1.000 | 0.996 | 0.711 | 0.987 | 0.897 | 85409 |
| | | TSDY_S | 1.000 | 0.996 | 0.459 | 0.995 | 0.913 | 39 |
| | Zinc | TSDY_B | 1.000 | 0.978 | 0.365 | 0.995 | 0.909 | 190 |
| | | TSDY_M | 1.000 | 0.978 | 0.688 | 0.996 | 0.916 | 5681 |
| | | TSDY_S | 1.000 | 0.978 | 0.393 | 0.997 | 0.939 | 18 |
| TSID | ChEMBL | TSID_B | 1.000 | 0.996 | 0.004 | 0.998 | 0.925 | 4267 |
| | Zinc | TSID_B | 1.000 | 0.978 | 0.010 | 0.999 | 0.945 | 190 |

The TSSA codes achieve the highest novelty scores, while the TSDY codes achieve lower scores and the TSID codes achieve almost zero. All codes receive reasonable FCD scores. "KLD" stands for Kullback–Leibler divergence. "FCD" represents Fréchet ChemNet Distance. "FBTs" refer to the types of Full Binary Trees in the training dataset. The suffix letters "J, B, M, S" in various t-SMILES code names represent the fragmentation algorithm: JTVAE, BRICS, MMPA, and Scaffold.

**Table 2 | Results on JNK3 active molecules using MolGPT with different training epochs**

| Model3 | Valid | Novelty | FCD | Active Novel | FBT Novel | Frag Novel |
|---|---|---|---|---|---|---|
| SMILES[R200] | 0.795 | 0.120 | 0.584 | 0.072 | N/A | N/A |
| SMILES[R2000] | 1.000 | 0.001 | 0.765 | 0.004 | N/A | N/A |
| DSMILES[R200] | 0.677 | 0.076 | 0.510 | 0.043 | N/A | N/A |
| DSMILES[R2000] | 0.999 | 0.001 | 0.778 | 0.001 | N/A | N/A |
| SELFIES[R200] | 1.000 | 0.238 | 0.544 | 0.148 | N/A | N/A |
| SELFIES[R2000] | 1.000 | 0.008 | 0.767 | 0.050 | N/A | N/A |
| TSSA_S[R300] | 1.000 | 0.833 | 0.564 | 0.582 | 2.655 | 0.962 |
| TSSA_S[R5000] | 1.000 | 0.817 | 0.608 | 0.564 | 2.534 | 0.049 |
| TSSA_S[R50000] | 1.000 | 0.824 | 0.572 | 0.571 | 2.379 | 0.023 |
| TSSA_HSV[R200] | 1.000 | 0.483 | 0.680 | 0.350 | 2.086 | 5.044 |
| TSSA_HSV[R2000] | 1.000 | 0.447 | 0.716 | 0.319 | 1.810 | 0.365 |
| TSSA_Hybrid[R200] | 1.000 | 0.683 | 0.622 | 0.374 | 2.310 | 25.978 |
| TSSA_Hybrid[R2000] | 1.000 | 0.657 | 0.619 | 0.437 | 2.672 | 23.745 |
| TF_SMILES[R5] | 0.887 | 0.707 | 0.523 | 0.526 | N/A | N/A |
| TF_SMILES[R100] | 0.999 | 0.033 | 0.764 | 0.023 | N/A | N/A |
| TF_TSSA_S[R5] | 1.000 | 0.932 | 0.483 | 0.710 | 2.897 | 9.105 |
| TF_TSSA_S[R100] | 1.000 | 0.849 | 0.570 | 0.569 | 2.431 | 0.208 |
| SMILES_Aug50[R10] | 0.807 | 0.570 | 0.566 | 0.483 | N/A | N/A |
| SMILES_Aug50[R100] | 0.995 | 0.049 | 0.750 | 0.047 | N/A | N/A |
| TSSA_S_Rec50[R10] | 1.000 | 0.962 | 0.389 | 0.829 | 2.414 | 1.757 |
| TSSA_S_Rec50[R100] | 1.000 | 0.960 | 0.411 | 0.809 | 2.448 | 0.655 |

"Active Novel" means the newly generated novelty molecules predicted by the AttentiveFP model as active. "FBT Novel" means different FBT (Full Binary Tree) compared with the training data. "Frag Novel" means different newly generated fragments compared with the training data. "R"means training epochs. "Aug" means augmenting training data by enumerating SMILES. "Rec" means reconstructing directly from active molecules to generate new active molecules as training data. "TF" means transfer learning. TSSA_HSV means hybrid model on TS_Vanilla and TSSA_S. TSSA_Hybrid means hybrid models on all TSSA codes including JTAVE, BRICS, MMPA, and scaffold-based TSSA and TS_Vanilla. See SI.D.3 and SI.D.4 for more detailed results.

datasets of structurally complex molecules, where even small amounts of data augmentation can have a negative impact on performance.

Due to the fact that the enumerated strings represent the same molecules, the data augmentation method implemented by enumerating SMILES cannot access a larger chemical space in the real chemical sense, but it is possible to provide more strings for the deep learning model to learn the SMILES syntax. Of course, the skills used to augment SMILES string can also be used to augment t-SMILES string at the fragment level. In addition, similar to the enumeration of SMILES, t-SMILES can be enumerated using different fragments as the root when generating FBT. Compared to SMILES, if we use reconstruction as a data augmentation method for t-SMILES, the later would generate different molecules from the same set of fragments. As a result, a broader space could be easily accessed in the chemical sense.

**Experiments on low-resource datasets**

The scarcity of labeled data presents a challenge for implementing deep learning in target-oriented drug discovery. This section simulates a real-world scenario, where novel compounds are discovered based on a small set of pre-existing bioactive compounds. This is because in the generic large compound libraries, such as Zinc, the vast majority of compounds have a very low probability to exhibit the desired bioactivity for a specific target protein. As a result, the chances to identify novel, high-quality leads from large compound repositories are low[43].

In addition, generative models that rely on distributional learning tend to learn the distribution of the training data. Furthermore, when limited training data is used, deep learning models are prone to overfitting, which can lead to decreased generalization performance. For these reasons, we only use active molecules as training data in this

section to investigate the overfitting problem and explore the active chemical space.

A discrimination model based on AttentiveFP[44] is trained to predict whether the newly generated molecules are active. For JNK3, the ROC_AUC (Area Under the Receiver Operating Characteristic) values of this AttentiveFP discrimination model are as follows: training: 0.995, validation: 0.980, and testing: 0.989.

We evaluate two low-resource datasets: JNK3 with 923 active molecules and AID1706 with only 329 active molecules. Please reference SI.D.1 and SI.D.11 for a comparison of active and inactive molecules. The figures indicate that AID1706 presents more challenges than JNK3. Nonetheless, TSSA_S models exhibit significantly higher novelty scores than SMILES. Please see SI.D.9–11 for the experimental results of AID1706.

**Overfitting problem on JNK3**

To investigate the overfitting problem against different algorithms on JNK3, we first train generative models with different training epochs for SMILES, DSMILES, SELFIES, and t-SMILES. In addition, data augmentation and pre-trained then fine-tuned models are also trained on SMILES and t-SMILES. See Table 2, Fig. 3 and SI.D for detailed experimental results.

Overall, the models based on SMILES, DSMILES, SELFIES, and t-SMILES show a consistent increase in FCD scores with the increasing number of training iterations. Notably, after the 200th training epoch, the FCD curves of SMILES, DSMILES, and SELFIES surpass that of t-SMILES in SI. Fig. 38 and subsequently stabilize independently. Additionally, the data presents a significant negative correlation between novelty and FCD scores.

In detail, the novelty scores of SMILES and DSMILES initially increase, but then sharply decline after 200 training epochs and eventually almost reach zero. SELFIES model experiences an abrupt drop in novelty score at the 200 epochs, also falling to almost zero. In contrast, t-SMILES's novelty score initially fluctuates slightly, but then stabilizes at a value of -0.8. The hybrid t-SMILES models surpass all t-SMILES models in FCD scores. Additionally, they achieve notably higher novelty scores and marginally lower FCD scores than the SMILES, DSMILES, and SELFIES models.

Regarding the notably high novelty scores during the 50th and 100th training epochs of SELFIES, this is because the SELFIES algorithm always transforms the generated strings into valid molecules. However, at these points, the FCD scores nearly approach zero, indicating that the newly produced molecules are vastly different from the training data. Refer to SI.D.7 for the variously produced molecules during differing training epochs.

Besides that, in transfer learning scenario, when the number of training epochs is increased from 5 to 100, the active novelty score of the t-SMILES-based model decreases from 0.710 to 0.569. However, the SMILES-based model experiences a drastic drop from 0.526 to 0.023.

When analyzing the models based on data augmentation, an increase in the number of training epochs from 10 to 100 leads to a striking decrease in active novelty score for the SMILES-based model: from 0.483 to 0.047. Nevertheless, the t-SMILES based models preserve their high levels of active novelty scores, reaching 0.829 and 0.809, respectively, with only slight decreases.

When evaluating newly generated active molecules, the typical TSSA_S model, data augmentation and transfer learning models obtain higher scores, which were 0.698, 0.829 and 0.710, respectively. On the contrary, the scores of the SMILES based model are 0.084, 0.483, and 0.526, respectively. The highest novel-active score of 0.829 comes from the t-SMILES-based model using reconstructed active molecules as training data.

Systematic experiments indicate that the special multiscale coding algorithm and reconstruction philosophy employed by t-SMILES

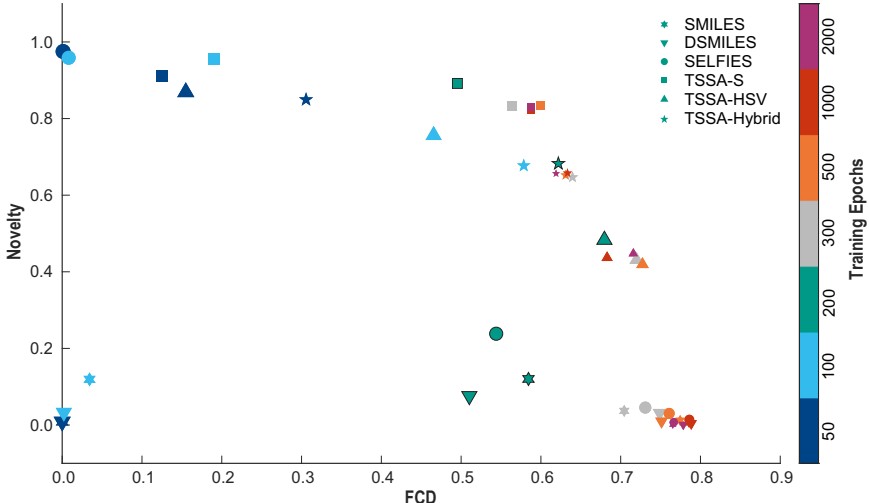

**Fig. 3 | Novelty and FCD (Fréchet ChemNet Distance) scores against SMILES, DSMILES, SELFIES, TSSA_S, TSSA_HSV, and TSSA_Hybrid on different training epochs.** Seven data points from large to small for each code indicate the number of training epochs:50, 100, 200, 300, 500, 1000, and 2000 respectively. The number of training epochs is indicated by the use of various sizes and gradient colors. TSSA_Hybrid represents the hybrid codes, including TS_Vanilla, TSSA_J (JTVAE-based TSSA), TSSA_B (BRICS-based TSSA), TSSA_M (MMPA based TSSA) and TSSA_S (Scaffold-based TSSA). In general, all models demonstrate a consistent increase in FCD scores with the increasing number of training iterations. After 200 training epochs, which are marked by a stroke, the novelty scores of the SMILES, DSMILES, and SELFIES models exhibit a notable decline, reaching a value close to zero. In contrast, the singleton or hybrid t-SMILES models exhibit considerably higher score stability, with values of over 0.8, 0.6, and 0.4 for TSSA_S, TSSA_HSV, and TSSA_Hybrid, respectively. The results indicate that t-SMILES models exhibit superior performance in comparison to SMILES, DSMILES, and SELFIES models. This is achieved by avoiding "striking similarity" to the training dataset and by achieving "better novelty with reasonable similarity".

effectively avoid the overfitting problem on low-resource datasets. To achieve a desired distribution of molecules in real-world experiments, selecting an appropriate molecular description and optimization parameters to balance FCD and novelty scores is essential.

The statistical data also suggests that utilizing data augmentation or pre-trained models could be more beneficial for SMILES model with limited resources. However, a standard TSSA_S model may achieve better results compared to models using SMILES-based data augmentation and transfer learning techniques. t-SMILES based transfer learning models achieved the highest scores among all models.

Furthermore, this experiment has proved that the t-SMILES algorithm can build a multi-code system for molecular description, in which each decomposition algorithm acts as a kind of chemical word segmentation algorithm splitting a molecule as different strings. All of these codes complement each other and contribute to a mixed chemical space. If one code is insufficient for inference, another code can provide complementary information. In this framework, classical SMILES can be unified as a special case of t-SMILES to achieve better-balanced performance using hybrid decomposition algorithms.

The t-SMILES models achieve higher novelty scores in a single round iteration in this experiment, which is a significant advantage. In goal-oriented tasks, we will provide additional evidence to demonstrate their iterative advantages.

In terms of physicochemical properties, SI.D.6 demonstrates that the SELFIES model requires further optimization to match the performance of SMILES.

In summary, experiments on JNK3 and AID1706 show that t-SMILES can avoid the overfitting problem and significantly improve the performance of balanced Novelty-FCD scores on labeled low-resource datasets, which is considered a major domain-specific challenge in molecular design.

### Experiments on ChEMBL
These experiments aim to evaluate the possibility of selecting and optimizing t-SMILES-based models to outperform SMILES, DSMILES, SELFIES, and baseline models on ChEMBL, especially in goal-directed tasks. Additionally, TSID and TSDY exhibit better performance in learning the distribution of the training data than TSSA on larger molecules.

### Distribution learning on ChEMBL
As shown in Table 3, t-SMILES based models outperform Graph MCTS, hgraph2graph (hG2G), and SOTA graph mode MGM. It seems difficult for Graph MCTS to capture the properties of the training data as shown by its almost-zero FCD score.

When comparing t-SMILES models with fragment-based assembling algorithm: FASMIFRA[45], all TSDY and TSID-based models outperform it in both novelty and FCD dimensions. However, this kind of assembly algorithm is expected to be used as a reconstruction process for the t-SMILES framework to improve performance in the future.

Compared to hG2G which is an advanced model based on model JTVAE[34] and aims to solve larger molecular problems with motif-based method, t-SMILES based modes have higher FCD scores and higher or similar novelty scores. Compared to MGM, TSSA-based models demonstrate higher novelty scores, but lower FCD scores. However, both TSDY and TSID models outperform the MGM model in both novelty and FCD.

In the realm of sequence-based baseline models, compared with AAE[46], t-SMILES models get higher FCD scores and lower novelty score. Compared to CharacterVAE[46], t-SMILES models get lower or similar novelty scores, and TSDY and TSID models achieve similar or higher FCD scores. The ORGAN[46] model shows poor performance in all valid, novelty, and FCD scores, indicating possible mode collapse. Regarding Transformer Reg[47], it receives lower novelty and FCD scores than MolGPT[4], this is mainly because they use different model structure, resulting in lower performance than some t-SMILES models.

When evaluating three baseline string representations (SMILES, DSMILES, and SELFIES), SMILES model obtains the highest FCD score of 0.906. DSMILES model receives lower scores for all five metrics when compared to SMILES. While the SELFIES model achieves the highest novelty score of 0.958, it is important to note that in our

**Table 3 | Results for the distribution-learning benchmarks on ChEMBL using GPT**

| | Model | Valid | Unique | Novelty | KLD | FCD | Nov./Uni. |
|---|---|---|---|---|---|---|---|
| Baseline Graph | Graph MCTS[46,56] | 1.000 | 1.000 | 0.994 | 0.522 | 0.015 | N/A |
| | hG2G[18] | 1.000 | 0.995 | 0.940 | 0.888 | 0.506 | N/A |
| | MGM[47] | 0.849 | 1.000 | 0.722 | 0.987 | 0.845 | N/A |
| Baseline SMILES | LSTM[40,46] | 0.959 | 1.000 | 0.912 | 0.991 | 0.913 | N/A |
| | CharacterVAE[2,46] | 0.870 | 0.999 | 0.974 | 0.982 | 0.863 | N/A |
| | AAE[46] | 0.822 | 1.000 | 0.998 | 0.886 | 0.529 | N/A |
| | ORGAN[3,46] | 0.379 | 0.841 | 0.687 | 0.267 | 0.000 | N/A |
| | Transformer Reg[31,47] | 0.961 | 1.000 | 0.846 | 0.977 | 0.883 | N/A |
| | MolGPT[4] | 0.981 | 0.998 | 1.000 | 0.992 | 0.907 | N/A |
| | FASMIFRA[45] | 1.000 | 0.994 | 0.702 | 0.959 | 0.814 | N/A |
| String | SMILES_[R10] | 0.980 | 0.979 | 0.907 | 0.992 | 0.906 | 0.926 |
| | DSMILES_[R10] | 0.898 | 0.897 | 0.836 | 0.989 | 0.893 | 0.933 |
| | DSMILES_[R15] | 0.910 | 0.908 | 0.845 | 0.992 | 0.896 | 0.930 |
| | SELFIES_[R10] | 1.000 | 1.000 | 0.958 | 0.979 | 0.857 | 0.959 |
| | SELFIES_[R15] | 1.000 | 0.999 | 0.953 | 0.983 | 0.865 | 0.954 |
| t-SMILES Family | TS_Vanilla_[R10] | 1.000 | 0.999 | 0.914 | 0.993 | 0.901 | 0.915 |
| | TS_Vanilla_[R15] | 1.000 | 0.998 | 0.907 | 0.994 | 0.907 | 0.909 |
| | TSSA_J_[R10] | 1.000 | 0.993 | 0.969 | 0.971 | 0.712 | 0.975 |
| | TSSA_B_[R20] | 1.000 | 0.995 | 0.956 | 0.972 | 0.708 | 0.961 |
| | TSSA_M_[R50] | 1.000 | 0.996 | 0.970 | 0.982 | 0.808 | 0.974 |
| | TSSA_S_[R50] | 1.000 | 0.998 | 0.977 | 0.966 | 0.795 | 0.979 |
| | TSSA_HJBMSV_[R20] | 1.000 | 0.998 | 0.970 | 0.964 | 0.825 | 0.971 |
| | TSDY_B_[R15] | 1.000 | 0.999 | 0.960 | 0.977 | 0.854 | 0.961 |
| | TSDY_M_[R15] | 1.000 | 0.998 | 0.970 | 0.960 | 0.852 | 0.972 |
| | TSDY_S_[R15] | 1.000 | 0.999 | 0.955 | 0.982 | 0.878 | 0.956 |
| | TSDY_HBV_[R15] | 1.000 | 0.999 | 0.943 | 0.988 | 0.897 | 0.944 |
| | TSDY_HMV_[R10] | 1.000 | 0.998 | 0.962 | 0.973 | 0.872 | 0.963 |
| | TSDY_HSV_[R10] | 1.000 | 0.999 | 0.950 | 0.985 | 0.891 | 0.951 |
| | TSDY_HBMSV_[R10] | 1.000 | 0.999 | 0.964 | 0.973 | 0.883 | 0.966 |
| | TSID_B_[R10] | 1.000 | 0.999 | 0.941 | 0.989 | 0.909 | 0.942 |
| | TSID_M_[R10] | 1.000 | 0.998 | 0.942 | 0.968 | 0.892 | 0.945 |
| | TSID_S_[R10] | 1.000 | 0.999 | 0.933 | 0.991 | 0.909 | 0.935 |
| | TSID_HBV_[R15] | 1.000 | 0.999 | 0.941 | 0.989 | 0.883 | 0.941 |
| | TSID_HBMSV_[R10] | 1.000 | 0.999 | 0.953 | 0.982 | 0.893 | 0.954 |

The results of ORGAN[3,46], LSTM[40,46], CharacterVAE[2,46], AAE[46] and Graph MCTS[46,56] are taken from GuacaMol[46], Transformer Reg[31,47] and MGM[47] are taken from ref. 47, MolGPT[4] is taken ref. 4, FASMIFRA[45] is taken from its reference, the results of hgraph2graph[18] is calculated by us. CReM[38] is not included as a baseline due to its nearly zero FCD score, even though its novelty score is close to 1. All other models are trained by us. Models based on TSSA, TSDY and TSID are trained in different epochs, with "R" indicating the number of training rounds, such as "[R10]". The letter "H" in t-SMILES code names indicates a hybrid model, while the letters "J", "B", "M", and "S" indicate fragmentation algorithm: JTVAE, BRICS, MMPA, and Scaffolds, "V" indicates TS_Vanilla code. "KLD" stands for Kullback–Leibler divergence. "FCD" represents Fréchet ChemNet Distance. "Nov./Uni." represents the ratio of a novelty score to a uniqueness score. Refer to SI Table 8 for repeatability.

previous discussion of the experiments on JNK3 and SI.D.7, we observed an especially high novelty score for SELFIES. This indicates a significant gap with the training data, which is supported by the lowest FCD score of 0.857.

Comparing three t-SMILES code algorithms, it is clear that TSDY and TSID have higher FCD and similar novelty scores compared to TSSA in this experiment. An in-depth investigation is encouraged to verify whether TSSA code algorithm could benefit from its distinctive structure, as some pieces are bonds that are broken. This could serve as a starting point for further research.

If evaluating t-SMILES and its alternatives, all tested t-SMILES models achieve higher novelty scores than SMILES model. If taking the SMILES FCD score of 0.906 as a standard and comparing the novelty score, the TSID_B and TSID_S model achieves an FCD score of 0.909 and a novelty score of 0.941 and 0.933, surpassing SMILES in both dimensions. When the TS_Vanilla model is trained for an additional five epochs, it achieves almost the same FCD score as SMILES, and a similar novelty score. Please ref to SI.B.3.3.1 for further analysis.

Considering the second high FCD score 0.896, which comes from DSMILES, and comparing novelty score, TSDY_HBV, TSID_B and TSID_S achieve both higher novelty and FCD scores. When the novelty score is set to 0.958, which is the highest of three alternatives, and the FCD score is compared, TSDY_HMV, and TSDY_HBMSV achieve both higher novelty and FCD scores.

Therefore, this experiment suggests that it is possible to select and optimize t-SMILES-based models to outperform SMILES, DSMILES, and SELFIES on ChEMBL.

**Physicochemical properties on ChEMBL**
The detailed results of physicochemical properties are summarized in Fig. 4a–c and SI.E.4.

Tables and figures demonstrate that the baseline model hG2G obtains the worst score in six out of nine categories, and the second highest score in another category. Further investigation reveals that hG2G produces more molecules with fewer atoms and rings than the training data, resulting in smaller values for MolWt. However, TSSA_J

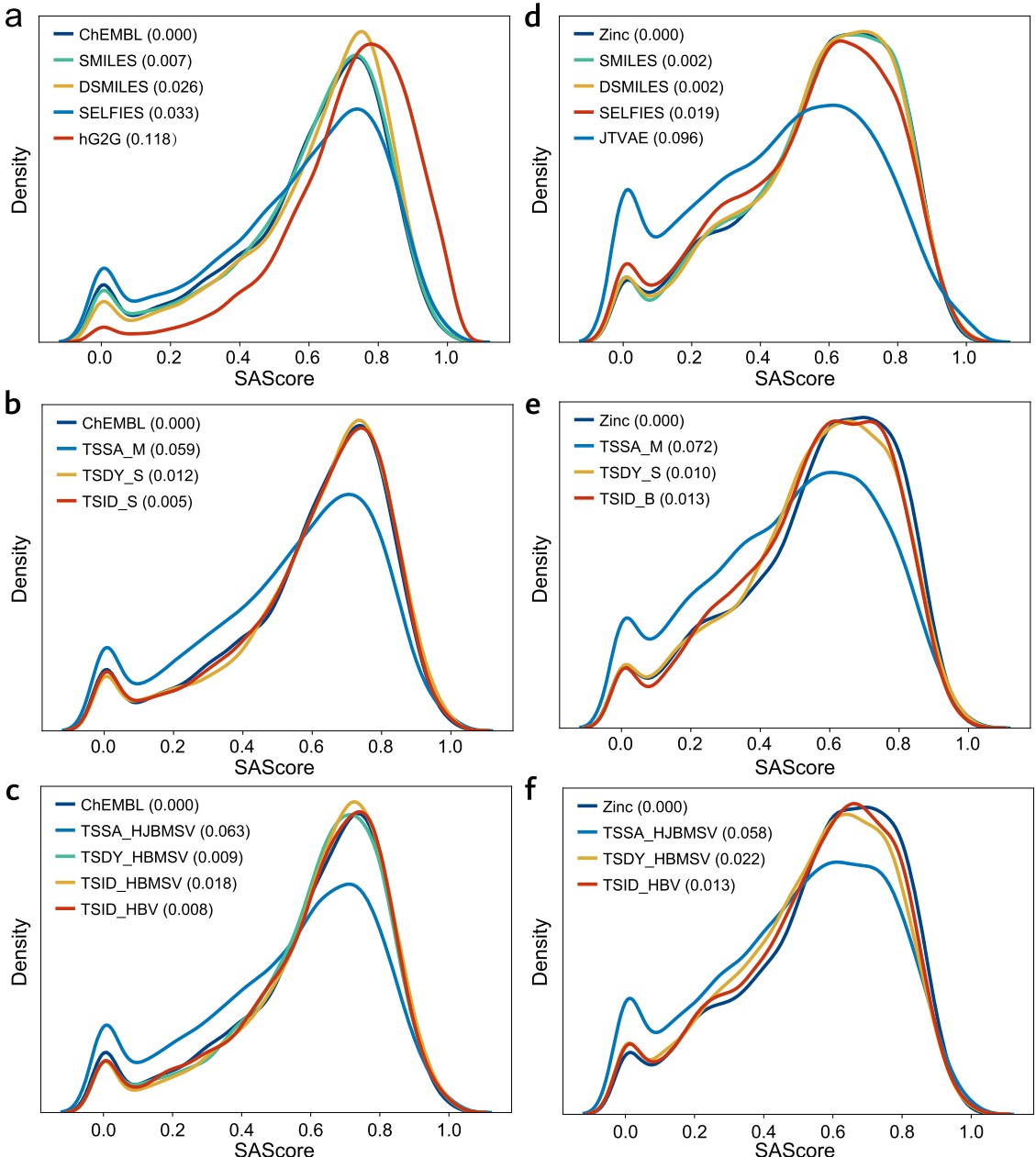

**Fig. 4 | Distribution of SAScore for training dataset and sets of generated molecules for ChEMBL and zinc.** For more detailed information on more curves and Wasserstein distance metrics, please refer to sections SI.E.4 and SI.E.5. **a, b** and **c** Baseline, Singleton t-SMILES and Hybrid t-SMILES models on ChEMBL. **d, e** and **f** Baseline, Singleton t-SMILES and Hybrid t-SMILES models on Zinc. The letter "H" indicates a hybrid model, while the letters "J", "B", "M", and "S" indicate fragmentation algorithms: JTVAE, BRICS, MMPA, and Scaffolds, respectively. "V" indicates

TS_Vanilla. Figures suggest that some singleton or hybrid models in the t-SMILES family seem to capture these physicochemical properties as effective or better than SMILES, DSMILES, and SELFIES in a similar experiment. Furthermore, TSDY and TSID models provide a much better fit to the training data compared to TSSA models. The baseline models hG2G and JTVAE demonstrate limited capabilities for pattern learning from the training data, with the lowest performance.

can match the training data almost perfectly in terms of N_Atoms and N_Rings, please refer to sections SI.E.4.1 and SI.E.4.2 for more information.

Compared to SMILES, both DSMILES and SELFIES models receive higher scores in fewer than four properties. While multiple t-SMILES models, especially TSID_S, outperform SMILES by more than four properties. This suggests that some singleton or hybrid models in the t-SMILES family seem to capture these physicochemical properties better than SMILES, DSMILES, and SELFIES in a similar experiment. Furthermore, the figures and tables indicate that TSDY and TSID models may provide a better fit to the training data compared to TSSA models on ChEMBL.

### Goal-directed learning
We first evaluate t-SMILES codes with a random reconstruction algorithm, using the same default settings as GuacaMol, on twenty subtasks. Please refer to SI.B.3.3.3 for comprehensive results. Further in-deep investigation indicates that models based on t-SMILES, as well as its substitutes, are not adequately trained for optimal performance when using the default settings for some sub-tasks. Consequently, this result serves as a starting point for further research with more training epochs and the goal-directed reconstruction algorithm.

Supplementary Table 13 and figures show that all models could get almost 1.0 scores on the first six subtasks. In addition, the results of BRICS, MMPA, and Scaffold based models suggest that different

fragmentation algorithms receive higher scores in different tasks. CReM uses a fragment-based framework that differs from a deep neural network model. It achieves the highest top-two score, with 14 out of 20 tasks. However, it performs poorly in sub-tasks 9, 19, and 20. SELFIES fails to reach any of the top-two highest ratings, while neither SMILES nor TSDY-Scaffold obtain the top-two lowest scores. DSMILES earns only one top-two highest ranking with six top-two lowest rankings.

A comparison of string-based representations shows that DSMILES and SELFIES require extra optimization to achieve performance comparable to SMILES. In contrast, the t-SMILES family yields diverse higher scores across different sub-tasks.

## Goal-directed reconstruction

As a crucial component of t-SMILES framework, besides the random assembly algorithm, we will evaluate three typical subtasks using goal-directed reconstruction algorithm to explore the limits of different codes, including "Median molecules 1" (T9.MM1), "Sitagliptin MPO" (T16.SMPO), and "Valsartan SMARTS" (T18.VS). For comparison, we increase the number of training epochs from 20 to 50 for T9.MM1 and T18.VS, and to 100 for T16.SMPO, for better scores. Detailed experiments are presented in Table 4, Fig. 5, and SI.B.3.4. CReM achieves the highest scores in SI.B.3.3.3 and serves as the standard in this in-depth evaluation.

The tables and figures show that all models achieve lower scores on T9.MM1 and higher scores on T18.VS. It seems that the target problems are one of the key factors determining whether there is a high scoring solution. For molecules with high scores, please refer to SI.B.3.7 and SI.B.3.8. Despite this, the t-SMILES models with goal-directed reconstruction algorithm still achieve the best performers for both two tasks.

Regarding T16.SMPO, additional experiments and analysis are conducted as presented in Fig. 5, Suppl. Fig. 10 and Suppl. Fig. 11. All six t-SMILES-based models significantly outperform SMILES-based models. In particular, among the models tested, TSDY_M models achieve significantly higher scores. These findings suggest that t-SMILES could have further applicability in reaching the target function's limits, and ultimately help chemists create more rational target functions. Please refer to SI.B.3.4 for further experiments and analyses.

Furthermore, from figures in SI.B.3.3.6 and SI.B.3.3.7 for T18.VS, one can see that the optimized molecules appear significantly different and their physicochemical properties span a wide range around the target properties. Meanwhile, there is a noteworthy difference among molecules with different scores in different codes regarding T16.SMPO, as shown in SI.B.3.3.9. Therefore, in practical experiments to ensure more favorable results in multi-objective optimization tasks, it is advised to use singleton or hybrid t-SMILES model with goal-directed reconstruction algorithm, which best suits the given task and is well optimized.

In summary, the three types of experiments on ChEMBL indicate that it is possible to select and train a t-SMILES based singleton or hybrid mode that significantly outperforms SMILES, DSMILES, SELFIES based models and some SOTA fragments and graph baseline models, especially in goal-directed tasks. In addition, TSDY and TSID-based

models achieve higher performance than TSSA-based models in capturing the distribution of the training data on larger molecules.

## Experiments on zinc

**Distribution learning.** As shown in Table 5, from the comparative analysis of baseline models, the t-SMILES-based models in this study significantly outperform the existing fragment-based baseline models. Model JTVAE[34] is one key baseline for this study. So far as the validity is concerned, both t-SMILES and JTVAE[34] models could generate almost 100% valid molecules. However, t-SMILES exhibits significantly higher FCD scores and similar novelty scores.

As to Group-VAE[25], which uses fragments and SELFIES, it achieves a lower novelty score compared to SELFIES-VAE in published experiments. Because Group-VAE uses a different way to calculate FCD, it is impossible to make a direct comparison. But in this experiment, SELFIES based model obtains lower scores for both novelty and FCD when compared to seven t-SMILES based models.

In terms of three string descriptions, the DSMILES model achieves the lowest validity and novelty scores. The SELFIES model is able to generate 100% valid molecules and gets higher novelty score than SMILES and DSMILES; however, it achieves the lowest FCD score.

To evaluate the t-SMILES family with its alternatives, if we fix the FCD score 0.942 of SMILES and compare the novelty score, TSDY_B, TSID_B and TSDY_HBMSV get slightly lower FCD scores and higher novelty scores. Considering the highest novelty score of 0.971, which comes from SELFIES, seven models TSDY_B, TSDY_S, TSDY_HB, TSDY_HSV, TSID_B, TSDY_HBMSV, TSID_B and TSID_HBV can get both higher FCD and novelty scores.

If only comparing novelty scores with SMILES, all tested models achieve higher scores. It means that t-SMILES models, with almost 100% validity and relatively high FCD scores, could improve novelty to explore a wider molecular space. That is to say, t-SMILES is an effective molecular representation for fragment-based molecular tasks on Zinc as well.

**Physicochemical properties of zinc.** Figure 4d−f and SI.E.5 show that none of the three baseline models achieve better scores than SMILES.

The key baseline mode, JTVAE[34], gets the worst scores in four out of nine categories. Further investigation shows that the JTVAE[34] model produces more molecules with fewer atoms and rings than the training data, resulting in smaller values for MolWt. However, TSSA_J was able to match the training data almost perfectly on N_Atoms and N-Rings, as shown in SI.E.5.1 and SI.E.5.3. This is similar to hG2G on ChEMBL. As a result, JTVAE[34] perform the worst in learning patterns from the training data.

On the contrary, multiple t-SMILES models as well as DSMILES and SELFIES, outperform SMILES models by more than four properties. Based on these results, some t-SMILES, DSMILES, and SELFIES models are able to capture these physiochemical properties as effectively as SMILES in these experiments. Conversely, JTVAE[34] shows limited capabilities.

In summary, both the experiments of distribution learning and physicochemical properties on Zinc yield a similar result with ChEMBL that it is possible to train a t-SMILES-based singleton or hybrid mode

## Table 4 | Results of goal-directed benchmarks on ChEMBL

| ID | BN | DSs | CReM | SMG | DSG | SFG | TSBR | TSBG | TSSR | TSSG | TSMR | TSMG |
|----|------|-------|-------|-------|-------|-------|-------|-------|-------|-------|-------|-------|
| 9 | MM1 | 0.297 | 0.360 | 0.453 | 0.383 | 0.410 | 0.446 | 0.421 | 0.443 | 0.452 | 0.438 | 0.467 |
| 16 | SMPO | 0.496 | 0.708 | 0.598 | 0.625 | 0.725 | 0.692 | 0.755 | 0.693 | 0.788 | 0.866 | 0.930 |
| 18 | VS | 0.258 | 0.987 | 0.985 | 0.988 | 0.986 | 0.991 | 0.987 | 0.994 | 0.997 | 0.985 | 0.996 |

For t-SMILES family, random(R) and goal-directed(G) reconstruction are evaluated. DSs means the scores of training data. "TS" in TSBR, TSBG, TSSR, TSSG, TSMR and TSMG means TSDY models. "B, M, S" in t-SMILES code names mean BRICS, MMPA and Scaffold based fragmentation algorithm. "SM" in SMG means SMILES model, "DS" in DSG means DSMILES model, "SF" in SFG means SELFIES model. "G" in SMG, DSG and SFG means more training rounds with 100. 10 or 15 random candidates are selected to calculate scores and the top-scoring one is chosen for output.

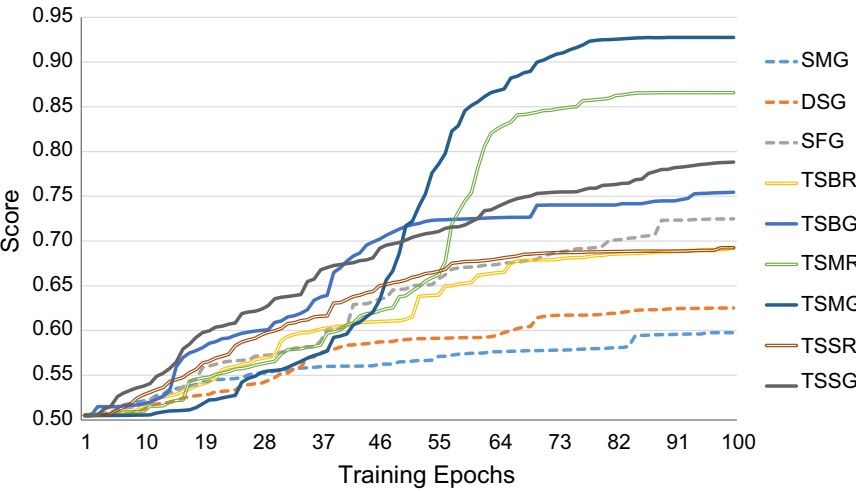

**Fig. 5 | Performance of the goal-directed benchmarks for T16.SMPO with different training epochs.** For t-SMILES family, random(R) and goal-directed(G) reconstruction are evaluated. "TS" in TSBR, TSBG, TSSR, TSSG, TSMR and TSMG means TSDY models. "B, M, S" in t-SMILES codes mean BRICS, MMPA and Scaffold based fragmentation algorithm. "SM" in SMG means SMILES model, "DS" in DSG means DSMILES model, "SF" in SFG means SELFIES model. "G" in SMG, DSG and SFG means more training rounds with 100. 10 or 15 random candidates are selected to calculate scores and the top-scoring one is chosen for output. The TSMG model yields significantly higher results. Further comprehensive experiments with different GPUs in SI.B.3.4 indicate that t-SMILES models exhibit high levels of repeatability with higher scores. Experiments against TSSA and TSID in SI.B.3.5 indicate that, similar to TSDY, TSSA models can outperform baseline models as well. This allows t-SMILES models to be used for the exploration of the limits in goal-oriented tasks.

**Table 5 | Results for the distribution-learning benchmarks on zinc using GPT**

|  | Model | Valid | Unique | Novelty | KLD | FCD | Nov./Uni. |
|---|---|---|---|---|---|---|---|
| Baseline Models | JTVAE[34] | 0.997 | 0.989 | 0.989 | 0.869 | 0.438 | 1.000 |
|  | FragDgm[23] | 1.000 | 0.518 | 0.517 | 0.824 | 0.311 | 0.997 |
|  | BRICS_Random | 1.000 | 1.000 | 1.000 | 0.959 | 0.656 | 1.000 |
|  | MolGPT[4] | 0.994 | 1.000 | 0.797 | N/A | N/A | N/A |
|  | SELFIES-VAE[25] | 1.000 | 0.999 | 0.735 | N/A | N/A | N/A |
|  | Group-VAE[25] | 1.000 | 0.999 | 0.719 | N/A | N/A | N/A |
| String | SMILES | 0.991 | 0.990 | 0.962 | 0.996 | 0.942 | 0.972 |
|  | DSMILES | 0.961 | 0.960 | 0.938 | 0.995 | 0.937 | 0.977 |
|  | SELFIES | 1.000 | 0.999 | 0.971 | 0.992 | 0.928 | 0.972 |
| t-SMILES Family | TS_Vanilla | 1.000 | 1.000 | 0.966 | 0.996 | 0.912 | 0.966 |
|  | TSSA_J | 1.000 | 0.996 | 0.992 | 0.987 | 0.832 | 0.996 |
|  | TSSA_B | 1.000 | 0.994 | 0.990 | 0.704 | 0.324 | 0.996 |
|  | TSSA_M | 1.000 | 0.996 | 0.990 | 0.988 | 0.861 | 0.994 |
|  | TSSA_S | 1.000 | 0.999 | 0.996 | 0.957 | 0.837 | 0.997 |
|  | TSSA_HJBMSV | 1.000 | 0.999 | 0.997 | 0.969 | 0.884 | 0.998 |
|  | TSDY_B | 1.000 | 1.000 | 0.974 | 0.993 | 0.933 | 0.974 |
|  | TSDY_M | 1.000 | 1.000 | 0.989 | 0.987 | 0.913 | 0.989 |
|  | TSDY_S | 1.000 | 0.999 | 0.983 | 0.993 | 0.929 | 0.983 |
|  | TSDY_HBV | 1.000 | 0.999 | 0.980 | 0.994 | 0.930 | 0.980 |
|  | TSDY_HMV | 1.000 | 0.999 | 0.990 | 0.990 | 0.923 | 0.990 |
|  | TSDY_HSV | 1.000 | 0.999 | 0.985 | 0.994 | 0.929 | 0.986 |
|  | TSDY_HBMSV | 1.000 | 0.999 | 0.991 | 0.985 | 0.932 | 0.992 |
|  | TSID_B | 1.000 | 0.999 | 0.976 | 0.992 | 0.932 | 0.976 |
|  | TSID_HBV | 1.000 | 1.000 | 0.983 | 0.993 | 0.929 | 0.983 |

MolGPT[4] is taken from ref. 4, SELFIES-VAE[25] and Group-VAE[25] are taken from ref. 25. All other models are trained by us. BRICS_Random model assembles BRICS-based fragments randomly. Models based on TSSA, TSDY and TSID are trained in the same number of epochs as SMILES. The letter "H" in t-SMILES code names indicates hybrid model, while the letters "J", "B", "M", and "S" indicate fragmentation algorithm: JTVAE, BRICS, MMPA, and Scaffolds, respectively, "V" indicates TS_Vanilla code. "KLD" stands for Kullback–Leibler divergence. "FCD" represents Fréchet ChemNet Distance. The term "Nov./Uni." represents the ratio of a novelty score to a uniqueness score.

that significantly outperforms baseline models on medium-sized molecules.

**Experiments on QM9.** On QM9, where the molecules are smaller, as illustrated in Suppl. Table 29 and Suppl. Fig. 63, all of the string-based models, including t-SMILES, SMILES, DSMILES, and SEFLIES except for CharacterVAE, achieve very high FCD scores close to or greater than 0.960, while having reasonably lower novelty scores. Our approach performs better than existing string-based approaches. Compared to graph-based baseline models, our approach demonstrates superior performance. For a comprehensive analysis of distribution learning and physicochemical properties metrics, please refer to SI.E.6.

## Discussion

When using advanced NLP methodologies to solve chemical problems, two fundamental questions arise: 1) What are "chemical words"? and 2) How can they be encoded as "chemical sentence"?

This study introduces a framework called t-SMILES to address the second question. The t-SMILES algorithm utilizes breadth-first search (BFS) to encode fragmented molecules as a string. Instead of using dictionary ID, classical SMILES is used to describe the fragments. This means that, t-SMILES can build a multiscale string-based molecular representation system, simplifying the computational complexity and enhancing the background knowledge, since the molecular structure is locally correlated in nature.

Systematic comparison experiments indicate that DSMILES and SELFIES require further optimization to match the performance of SMILES on some tasks due to their advanced grammar, as summarized by Krenn et al. In contrast, compared to classical SMILES, t-SMILES introduces only two additional symbols, "&" and "^", to encode multiscale and hierarchical molecular topology. Due to their resemblances, it is relatively straightforward to optimize t-SMILES models to outperform classical SMILES, especially TSDY and TSID.

Three code algorithms are presented in this work: TSSA, TSDY, and TSID. TSSA is recommended as the first choice for goal-directed tasks, especially for one-round low-resource tasks, to avoid "striking similarity" to the training dataset and achieve "better novelty with reasonable similarity". TSID is recommended for distribution reproduction experiments due to its accurate fit to the physicochemical properties of the training data. TSDY codes are a balanced choice for both goal-oriented and distributional reproduction tasks. Furthermore, both random and goal-oriented reconstruction algorithms are evaluated. The latter significantly outperforms SOTA baseline models in goal-directed tasks. In addition, t-SMILES offer a wider range of advantages, especially for low-resource datasets, achieving higher scores to balance novelty and FCD.

Meanwhile, the t-SMILES framework has the ability to unify classical SMILES as TS_Vanilla, making t-SMILES a superset of SMILES to build a multi-code molecular description system. In real-world molecular modeling tasks, medicinal chemists typically face challenging multi-objective optimization problems, with far more potential choices than can be systematically explored. Therefore, it is recommended to use hybrid t-SMILES approaches to explore as much of the chemical space as possible or a singleton t-SMILES model for a specific task. Moreover, t-SMILES makes it possible to generate valid and chemically diverse fragments when generating molecules, which can be served as foundation for other fragment-based researches.

### Scalability

Theoretically, t-SMILES algorithm is capable of supporting any effective substructure types and patterns as long as they can be used to obtain chemically valid molecular fragments. With the invaluable accumulation of drug fragments by experienced chemists, t-SMILES algorithm can combine this experience with a powerful sequence-based AI models to help chemists better exploring their experimental space.

In addition, t-SMILES is an open and flexible framework for encoding fragmented molecules using tree-based algorithms. If we use SMILES to encode the fragments, we get the t-SMILES string. If we use DSMILES or SELFIES to encode the fragments, we would get the t-DSMILES or t-SELFIES strings.

In some special cases, the t-SMILES algorithm could encode tree nodes by dictionary id instead of SMILES if only a specific fragment space needs to be explored and no expansion is required. Alternatively, we could replace the newly generated fragments with the fragments from the training data according to a given set of rules. In these cases, the coding logic and the flow of the algorithm of t-SMILES will remain unchanged.

### Limitations and to be improved

Large Language Models (LLMs) have recently demonstrated impressive reasoning abilities in various tasks. And some research has shown that LLMs can understand well-formed English syntax. So, whether the tree structure of t-SMILES could be learned and how LMs go beyond superficial statistical correlations to learn the chemical knowledge of molecules remains to be explored in depth.

This work focused on encoding fragmented molecules as sequence, so only published fragmentation algorithms are used as examples to create "chemical words". Future research could utilize t-SMILES to explore additional fragmentation algorithms and decipher chemical sentences and meanings more deeply, which is actually more challenging than NLP.

Although t-SMILES aims to improve the performance of molecule description and circumvent the limitations of SMILES, experiments on more complex molecules were not performed in this study. This will be a topic for future research.

Finally, this is a promising start for encoding fragmented molecules as SMILES-type strings. Further research could explore advanced algorithms for molecule reconstruction and optimization, improved generative models, and evolutionary techniques. Additionally, research could focus on property, retrosynthesis, and reaction prediction tasks.

## Methods

To support the t-SMILES algorithm, we introduce a new character, "&", to act as a tree node when the node is not a real fragment in FBT. Additionally, we introduce another new character, "^", to separate two adjacent substructure segments in t-SMILES string, similar to the blank space in English sentences that separates two words.

In this section, we begin by outlining the fundamental concept of t-SMILES. Subsequently, we highlight FBT and AMT, which constitute the key components of the t-SMILES algorithm. Next, we briefly introduce molecular fragmentation algorithms and molecular reconstruction strategy.

### t-SMILES algorithm overview

In the t-SMILES algorithm, a molecular graph is firstly divided into valid chemical fragments (or substructures, clusters, subgroups, subgraphs) using one specified disconnection method or some hybrid disconnection methods to obtain its AMT, as shown in Fig. 1. Following with the AMT being transformed into a FBT, and finally the FBT is traversed in breadth-first search (BFS) to obtain the t-SMILES string. During the reconstruction, the reverse process is used, and finally the molecular fragments are assembled into the chemical correct molecular graph.

### Algorithm steps to construct t-SMILES from a molecule

Step 1: Break down a molecule according to the selected molecular fragmentation algorithm to build AMT;
Step 2: Convert the AMT to a FBT;
Step 3: Traverse the FBT with BFS algorithm to get t-SMILES.

The BFS algorithm for the tree is a level-order traversal of tree. For any node w in the BFS tree rooted at v, the tree path from v to w in the tree corresponds to a shortest path from v to w in the corresponding graph. Please refer to SI.A.2 for a demonstration of the BFS algorithm.

Three coding algorithms are presented in this study:

1. TSSA: t-SMILES with shared atom.
2. TSDY: t-SMILES with dummy atom but without ID.
3. TSID: t-SMILES with ID and dummy atom.

The distinction between TSSA and TSDY/TSID is that in TSSA, two fragments share a real atom as a connecting point, whereas TSDY and

TSID use a dummy atom (indicated by character "*" with or without ID) to illustrate how the group bonds. TSDY and TSID have a simpler fragmentation and assembling logic compared to TSSA, as can be seen in Fig. 1 and SI Fig. 1. For additional examples and more detailed information on the various code algorithms, please refer to SI.A.1.

For example, in Fig. 1 and SI Fig. 1, the three t-SMILES codes of Celecoxib are:

1. TSID_M (Fig. 1): [1*]C&[1*]C1 = CC = C([2*])C = C1&[2*]C1 = CC([3*]) = NN1[5*]&[3*]C([4*])(F)F&[4*]F^[5*]C1 = CC = C([6*])C = C1& &[6*]S(N)( = O) = O& & &
2. TSDY_M (Fig. 1, replace [n*] with *): *C&*C1 = CC = C(*)C = C1&*C1 = CC(*) = NN1*&*C(*)(F)F&*F^*C1 = CC = C(*)C = C1& &*S(N)( = O) = O& & &
3. TSSA_M (SI. Fig. 1): CC&C1 = CC = CC = C1&CC&C1 = C[NH]N = C1&CN&C1 = CC = CC = C1^CC^CS&C&N[SH]( = O) = O&CF& & & & FCF& &

**Full binary tree**
A FBT is a special type of binary tree in which every parent node/internal node has either two or no children. As the most trivial tree, FBT's structure is regular and easy to calculate. The reason for using FBT with some redundant nodes instead of complete binary tree or other trees is that its algorithm and structure being easy to learn by deep learning models, and the redundant nodes could be used as global marker nodes. In this work, the character "&" (tree node marked as "&") marks the end of the tree branch in the FBT, capable of providing the global structural information describing the molecular topology in the t-SMILES string.

With chemically meaningful molecular fragmentation representation using FBT, t-SMILES effectively reduces the nesting depth of brackets in SMILES codes, weakens the long-term dependencies in grammar, and fundamentally reduces the difficulty of learning molecular information using sequence-based deep learning models. In t-SMILES algorithm, except for the extra two characters "&" and "^", no more symbols are introduced, nor are recursion and other sophisticated calculations with high computational complexity introduced.

**Molecule decomposition**
According to ref. 48, there exist four major strategies to fragment molecules: knowledge-based, synthetically oriented, random, and systematic or hierarchical. The open-source molecular toolkit RDKit[49] has implemented some molecular decomposition methods, such as BRICS[35], MMPA[36], and Scaffold[37], etc. Another molecular fragmentation algorithm in this study can find its root in feature tree[50] published for molecular similarity algorithm by Rarey et al. in 1998 and JTVAE[34] proposed later by Jin et al.

The first step in the t-SMILES algorithm involves decomposing molecules into valid chemical fragments, utilizing a specified disconnection algorithm. And then generating a reduced graph[51] according to the cutting-off logic of fragmentation algorithms and then calculating one of its spanning tree as AMT.

**Acyclic Molecular Tree**
The idea of using tree as the base data structure of algorithms to address molecular related tasks has been long established in cheminformatics. In early study of molecular descriptor and similarity analysis, algorithms such as reduced graph, feature tree[50,52] not only had shown potential power to improve the similarity search but also being capable of retrieving more diverse active compounds than using Daylight fingerprints[53]. Some recent works[54,55] proposed to incorporate tree-based deep learning models into molecular generation and synthesis tasks as well.

AMT being capable to describe the molecule at various levels of resolution, reduced graph provides summary representations of chemical structures by collapsing groups of connected atoms into single nodes while preserving the topology of the original structures. In reduced graph, the nodes represent groups of atoms and the edges represent the physical bonds between the groups. Constructing reduced graph in this way forms a hierarchical graph, whose top layer being the molecular topology representing global information, and the bottom layer representing molecular fragments of detailed information. Groups of atoms are clustered into a node in the reduced graph approach, which could be done based on fragmentation algorithms. The feature tree[50] is a representation of a molecule similar to a reduced graph. The vertices of the feature tree are molecular fragments and edges connect vertices that represent fragments connected in the simple molecular graph. In t-SMILES algorithm, the minimum spanning tree of the reduced graph and the concept of feature tree could be regarded as an AMT, and then the next encoding algorithm is done based on this AMT.

Specific to our experiments, one part of the approach is to generate AMT based on the tree logic fragmented by the BRICS, MMPA, and Scaffold. Another method uses a junction tree introduced by Jin et al. in JTVAE[34] as the AMT.

**Molecular reconstruction**
We follow the following steps to reconstruct molecules from t-SMILES strings.

Step 1: Decompose t-SMILES string to reconstruct the FBT;
Step 2: Convert the FBT to the AMT;
Step 3: Assemble molecular fragments according to the selected algorithm to generate the correct molecular graph and then optimize it.

During reconstruction, one key problem is how to assemble the molecular fragments together to get a "chemically correct" molecule. In this study, for efficiency reasons, we assemble molecular graph one neighborhood at a time, following the order in which the tree itself was generated. In other words, we start from the root node of AMT and its neighbors, then we proceed to assemble the neighbors and their associated clusters, and so on. If there is more than one candidate when assembling two pieces, we select one molecule based on some specific rules, such as randomly or using a goal-directed function defined in GuacaMol[46].

Assembling fragments to create a chemically valid molecule is a challenging task with a significant impact on the quality of molecules. Some algorithms such as Monte Carlo tree search, CReM[38], FASMIFRA[45] and eSynth[43] can serve as a starting points for future research in this field.

**Reporting summary**
Further information on research design is available in the Nature Portfolio Reporting Summary linked to this article.

## Data availability
The data that support the findings of this study are publicly available and can be accessed via the GitHub repository and Zenodo deposition with doi: Wu et al. t-SMILES: a fragment-based molecular representation framework for de novo ligand design, https://github.com/juanniwu/t-SMILES, 2024 Wu et al. t-SMILES: a fragment-based molecular representation framework for de novo ligand design. https://doi.org/10.5281/ZENODO.10991703, 2024 In addition, Source data are provided with this paper.

## Code availability
Code, training and generation scripts for this work can be found at the GitHub repository and Zenodo deposition with DOI: Wu et al. t-SMILES: A Fragment-based Molecular Representation Framework for De Novo Ligand Design, https://github.com/juanniwu/t-SMILES, 2024 Wu et al. t-SMILES: A Fragment-based Molecular Representation Framework for

De Novo Ligand Design. https://doi.org/10.5281/ZENODO.10991703, 2024 In addition, the code in this article has been certified as Reproducible by Code Ocean: Wu et al. t-SMILES: A Scalable Fragment-based Molecular Representation Framework for De Novo Molecule Generation, https://codeocean.com/capsule/3034546/tree, 2024.

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

## Acknowledgements
This research was funded by the National Natural Science Foundation of China including Nos. 21874040, 22174036 and 22204049.

## Author contributions
J.W. and R.Y. designed the study and manuscript. As the main designer of the project, J.W. conceived the project, constructed the algorithms and python script, performed the experiments, informatics analyses, and wrote the draft manuscript. T.W. and Y.C. participated in the experiments. L.T. and H.W. participated in the discussion and funding acquisition. All authors contributed to manuscript editing, revising and have approved the final version of the manuscript.

## Competing interests
The authors declare no competing interests.
