## [Peer Review File · Nature Communications]

t-SMILES: A Fragment-based Molecular Representation Framework for De Novo Ligand DesignREVIEWER COMMENTS

Reviewer #1 (Remarks to the Author):

Major comments:

=====

This manuscript is between two to four times too long. As a result, I don't understand what are the key contributions and key experiments that the authors would like to show us.

Maybe, the authors should just concentrate on t-SMILES and the distribution reproduction experiments from the GuacaMol benchmark. Dump all the rest, or try to publish the rest in a future paper, once t-SMILES have been established.

If indeed the authors have created a new string-based representation for molecules, then this is a key and much needed contribution nowadays. Please distill this paper and concentrate on t-SMILES Vs. SMILES, DeepSMILES and SELFIES. If you indeed improve on those three, or even just over SMILES and DeepSMILES, that would be interesting already.

Minor comments:

=====

- p5: why the validity of generated molecules is not 100%? Can't you shoot for this? You are at 99% already! If you have a proper algorithm, I don't see why it would not work all the time.

- at this step, potential users will only be interested by an open-source encoder and decoder for t-SMILES; possibly in Python

- p7 and Fig 1: I don't even understand how you fragment molecules. Can you show several famous molecules fragmented / t-SMILES encoded by your approach. caffeine, aspirin, paracetamol, etc. From a single example, and if your explanations are not good enough, how are people supposed to follow what you are doing? Humans, like deep-learning models, might need a few examples in order to generalize. The fragmentation in Fig 1 seems like heavy atoms were saturated with hydrogens after bonds were cut; some bonds were duplicated after breaking in two a fused ring system (what fragmentation algorithm on earth does this? and how to reconnect fragments after such a molecular "disintegration"?)

- are there constraints on the molecular fragmentation scheme t-SMILES can use? E.g. is opening rings OK?

- p15: proper data augmentation for SMILES is via SMILES

randomization; cf.

Randomized SMILES strings improve the quality of molecular generative models. *J Cheminform* 11, 71 (2019).

<https://doi.org/10.1186/s13321-019-0393-0>

Is this what you call SMILES enumeration?

Reviewer #2 (Remarks to the Author):

In this paper, the authors introduce t-SMILES, a new molecular representation that includes two additional special characters to the SMILES vocabulary to account for rings and branches in a way that does not require paired brackets or SELFIES tokenization. t-SMILES uses breadth-first rather than the depth-first search used for constructing SMILES, to reduce the nesting depth of characters and reduce the need for generative models to capture long-range dependencies in the grammar. They show benefits to generating valid and novel molecules when training generative models on this input representation.

Comments and questions

In line 259, it's stated that candidates are randomly selected when assembling pieces during reconstruction of a molecule from t-SMILES. What is the "recovery rate" for a common dataset (e.g. Zinc 250k) when converting back and forth from SMILES to t-SMILES? The inverse of this question may be answered in Table 2.

In line 288, are the authors saying that "&^" characters occur frequently enough in the tokenization for generative models to pick up on?

Because the authors refer to t-SMILES in the tables using the names of the approaches they use for substructure identification (e.g., JTVAE), I'd suggest highlighting those columns and somehow indicating that they are all flavors of t-SMILES.

In line 325, what is meant by "models that do not require training...introduce other problems such as long training time" ?

How does t-SMILES compare to CREM and Group SELFIES? These approaches also introduce chemical diversity through either "chemically reasonable mutations" (CREM) or assembling fragments (Group SELFIES).

In Fig 6 you might plot Novelty against FCD score and use colors and shapes to indicate methods and training epoch, to better show the tradeoff.

The t-SNE plots in Fig 7 do not have any clearly identifiable clustering structure and may be better suited to the SI if the conclusion is just that training and generated samples overlap.

Dear Reviewers,

Thank you very much for your valuable and helpful comments and suggestions which have helped us strengthen our manuscript. Based on your review, we have revised our manuscript and conducted additional investigations. We list the major changes below:

- 1) In response to the *Open-Ring* issue identified by *Reviewer 1* and the *comparison with CReM on ChEMBL* highlighted by *Reviewer 2*, we have added two additional t-SMILES code algorithms to the revised manuscript for better performance on larger molecules. This brings the total number of t-SMILES code algorithms to three: TSSA (t-SMILES with shared atom), TSDY (t-SMILES with dummy atom but without ID), and TSID (t-SMILES with ID of dummy atom).
- 2) In response to the *fragmentation procedure* issue raised by *Reviewer 1*, Fig.1 has been revised. In addition, more examples and SI.Fig.A.1.1. with nuanced comments have been added.
- 3) In response to the *comparison with CReM* identified by *Reviewer 2*, we conducted *Goal-Directed Learning* with 20 sub-tasks which is defined in GuacaMol benchmark using TSDY.
- 4) In response to the concern of *excessive length* highlighted by *Reviewer 1*, some sections and content have been condensed, merged, relocated, or removed.

We have conducted additional experiments, benchmarks and introduced more nuanced discussions in relevant sections. Furthermore, we revamped our conclusion to scrutinize the constraints of the present framework and identify potential prospects for future studies. We address all concerns as they arise during the review process and provide a detailed point-by-point response to these comments.

Response to comments of Reviewer 1 starts here and ends on page 10.

Response to comments of Reviewer 2 can be found in pages 11-15.

REVIEWER COMMENTS

Reviewer #1 (Remarks to the Author):

Major comments:

=====

This manuscript is between **two to four times too long**.

Thank you for expressing your concerns. We acknowledge that it would be very difficult to convince the audience to accept our proposal to make any changes to the famous classical SMILES system, particularly since we only proposed to introduces two new symbols, "&" and "^" for encoding multi-scale and hierarchical molecular topologies. The use of classical SMILES-type

strings is considered the most suitable method for describing molecules when borrowing advanced NLP methodologies to solve chemical problems. However, there are still some issues that need to be addressed and proven.

Actually, we really hoped to show the generality of our proposed approach as efficiently as possible by comparing it with SOTA tools. We apologize for not achieving our objective. Due to our limited experience, we submitted an original manuscript that did not meet the desired standards. However, we have done a major revision to enhance its academic quality. As listed in Response 1.3, we have condensed, merged, relocated, or removed some sections and content.

As a result, I don't understand what are the **key contributions and key experiments** that the authors would like to show us.

Response 1.0 Key contributions

Reviewer 2 has provided a clear and objective summary of contributions. It is quoted directly here:

In this paper, the authors introduce t-SMILES, a new molecular representation that includes two additional special characters to the SMILES vocabulary to account for rings and branches in a way that does not require paired brackets or SELFIES tokenization. t-SMILES uses breadth-first rather than the depth-first search used for constructing SMILES, to reduce the nesting depth of characters and reduce the need for generative models to capture long-range dependencies in the grammar.

They show benefits to generating valid and novel molecules when training generative models on this input representation.

Jean-Marie Lehn's famous quotation "Atoms are letters, molecules are the words, supramolecular entities are the sentences and the chapters"(Lehn, 1988)(Jean-Marie Lehn - Interview, n.d.) was cited by the researchers(Cadeddu et al., 2014) studying the rank distribution of fragments in organic molecules being similar to that of words in the English language. This idea inspired us to use advanced NLP methodologies for molecular modeling.

Therefore, we must first address two key questions: 1) What are 'chemical words' and 2) How can they be encoded as 'chemical sentence'?

Defining 'chemical words' or 'chemical fragments' is a significant challenging task, more difficult than word segmentation algorithms in NLP. Fortunately, there are some published algorithms available that can generate chemical fragments.

The main contribution of our work is the t-SMILES framework, along with its encoding and decoding algorithms, which address the second question. Since it is not possible to cover all ideas, research and experiments in one paper, as far as details are concerned, at least the following points should be included:

- 1) t-SMILES serves as a *scalable molecular description* to encode *fragmented* molecules as a *string*. Instead of using a dictionary ID, classical SMILES is utilized to describe the molecule fragments. As is widely recognized, SMILES somehow is more advantageous than Graph in describing molecules, such as chirality information. By utilizing t-SMILES, we can not only maintain SMILES' benefits but also enhance it.
- 2) Compared to atom-based classical SMILES, t-SMILES introduces *only two extra, unpaired* characters and using *BFS* than DFS algorithm. The main reason that classical SMILES model generates invalid strings is due to the deep nesting and the long-term dependencies. t-SMILES algorithm effectively *reduces the nesting depth* of characters, and the need for generative models to capture long-term dependencies in the grammar. On the other hand, the BFS algorithm obtains the shortest path between two nodes, which better reflects the properties of real molecules.
- 3) The t-SMILES framework includes a *decoding procedure* that reconstructs t-SMILES strings into valid chemical molecules. This integration unifies distributional and non-distributional approaches into a single system. Our research evaluated the random and goal-oriented algorithms separately and obtained promising results. Advanced algorithms like MCTS, CReM have the potential to further enhance performance in the future.
- 4) The t-SMILES is an *open and flexible framework*, which is able to integrate classical SMILES as a special case and construct a *multi-code system* for describing molecules that enables efficient exploration of larger chemical spaces. Our study has evaluated the JTVAE, BRICS, MMPA, Scaffold, and Open-Ring algorithms and demonstrated their diversity especially in low-resource and goal-oriented tasks. It is believed that t-SMILES will be able to support additional fragmentation algorithms, such as pharmacophore, functional group, maximum common substructures (MCS), etc., in the future to help address the challenges of molecular design tasks in real-world environments, and the research of the ‘chemical word’ problem.

On the other hand, although SMILES is used in this study to demonstrate the t-SMILES framework, it is important to note that other formats such as DeepSMILES or SELFIES can also be used to describe fragments, resulting in t-DeepSMILES or t-SELFIES, to take advantage of their benefits. This could be an interesting topic for further research.

5) t-SMILES uses *tree* to encode multi-scale and hierarchical molecular topologies, so whether the tree structure could be learned and how LMs go beyond superficial statistical correlations to learn the chemical knowledge of molecules remains to be explored in depth, since some research has shown that LLMs can understand well-formed English syntax.

Systematic experiments indicate that t-SMILES exhibits impressive performance on low-resource datasets, whether the model is original, data augmented, or pre-training fine-tuned. It significantly outperforms classical SMILES, DeepSMILES, and SELFIES in goal-directed tasks. Meanwhile, t-SMILES models surpass SOTA fragment, graph and string-based approaches on ChEMBL, Zinc, and QM9.

Response 1.1 Key experiments

Since well-optimized language models have been shown to be effective in various studies, including the open-ring problem and the goal-directed tasks, we evaluate the t-SMILES from some perspectives as follows:

Firstly, t-SMILES was systematically evaluated by exploring its distinguished properties, considering that the boundaries between different codes depend heavily on their fundamental distinctions. Following this, comprehensive experiments are performed on two labeled, low-resource datasets: JNK3 and AID1706. Our study aims to compare and evaluate the advanced benefits of t-SMILES and its alternatives, which were achieved through the use of standard, data augmentation, and pre-training fine-tuning models. In line with our goal, we evaluated twenty goal-directed tasks on ChEMBL in parallel, as presented in the revised manuscript.

Additionally, we conducted experiments on three widely-used datasets, ChEMBL, Zinc, and QM9, employing all code algorithms to evaluate the overall performance of t-SMILES. We evaluate t-SMILES by comparing it to its counterparts, the fragment-based and graph-based baseline models.

To evaluate the adaptability and flexibility of t-SMILES to fragmentation algorithms, we employed four previously published fragmentation algorithms (JVATE, BRICS, MMPA, and Scaffold) to break down molecules.

Furthermore, we conducted rapid ablation study in order to enhance our comprehension and confidence in t-SMILES. This experiment indicated that TSSA models get higher novelty score on all these models with reasonable FCD scores.

Below table is a brief summary of the experiments conducted in our study.

Code	Joint Point	Frag Alg.	Experiments	Relative Pros and Cons of TSSA, TSDY and TSID
TSSA	Shared Atom	JTVAE, BRICS, MMPA, Scaffold, Hybrid	ChEMBL Zinc QM9 Low-Resource GPT, LSTM Transfer Learning Data Augmented	 1. Highest reconstruction novelty score; 2. Generative model without training; 3. In-depth investigation is encouraged to verify whether TSSA code algorithm could benefit from its distinctive structure, as some pieces are bonds that are broken.
TSDY	Dummy Atom, without ID	BRICS, MMPA, Scaffold, Hybrid	ChEMBL Zinc QM9 Goal-Oriented GPT, LSTM	 1. Medium reconstruction novelty score; 2. Generative model without training; 3. High computational performance on reconstruction; 4. Better fitting training data on physicochemical properties; 5. Larger molecules;
TSID	Dummy Atom with ID	BRICS, MMPA, Scaffold, Open-Ring, Hybrid	ChEMBL Zinc QM9 Open-Ring GPT	 1. Almost zero reconstruction novelty score; 2. Need to be trained to build generative model; 3. High computational performance on reconstruction; 4. Better fitting training data on physicochemical properties; 5. Larger molecules; 6. Longest;
TS_Vanilla			ChEMBL Zinc QM9 Low-Resource Goal-Oriented GPT, LSTM	 1. Classical SMILS in t-SMILES format

This table has also been added to SI file as SI.Table.B.1.

Maybe, the authors should just concentrate on t-SMILES and the **distribution** reproduction experiments from the **GuacaMol benchmark**.

Response 1.2 GuacaMol benchmark

We are most grateful for your suggestion.

Overall, we use distribution-learning benchmarks which are described in GuacaMol to evaluate the fundamental performance of t-SMILES models. From the perspective of optimization, it could be considered that the task can be solved once the molecules with a high score of a certain index are generated, but these generated molecules may not be useful. Therefore, we also use Wasserstein distance metrics for physicochemical properties in various experiments to assess the ability of generative models to effectively learn the physical and chemical characteristics of molecules in the training set.

The Distribution-Learning metrics of models: ORGAN, LSTM, CharacterVAE, AAE, and Graph MCTS were listed as key baseline models in the original manuscript, with the exception of the random sampler. We have updated the reference information in the revised version to prevent any misleading reference numbers.

GuacaMol includes Distribution-Learning Benchmarks and Goal-Directed Benchmarks. In response to Reviewer 2, we performed goal-directed learning with 20 subtasks on ChEMBL using TSDY to evaluate t-SMILES, SMILES, DeepSMILES, and SELFIES in revised manuscript.

Dump all the rest, or try to publish the rest in a **future paper**, once t-SMILES have been established.

If indeed the authors have created a new string-based representation for molecules, then this is a key and much needed contribution nowadays.

Please **distill** this paper and concentrate on **t-SMILES Vs. SMILES, DeepSMILES and SELFIES**.

Response 1.3 Distil & future paper

We are very grateful for your review and suggestion. We have refined this paper to improve its academic quality. However, to ensure systematic evaluation, we have to retain some contents of the original version. Here is a list of condensed, merged, relocated and removed contents:

- 1) Condense ABSTRACT, Introduction and Conclusion.
- 2) Condense Properties
- 3) Condense section 3.2 Generative model without training.
- 4) Condense section 3.3 Chemical space and data augmentation for t-SMILES.
- 5) Condense and merge section 3.4 and 3.5: Experiments on JNK3.
- 6) Condense section 4.2.1, 4.2.2 and 4.2.3: Experiments on ChEMBL, Zinc and QM9
- 7) Relocate section 4.1.2 to S.I.D.9: t-SMILES on smaller dataset AID1706.
- 8) Relocate part of experiments on QM9 to S.I.E.6.
- 9) Remove section 3.1.2 Entropy.
- 10) Remove section 4.3 LSTM, Transformer, VAE and AAE for t-SMILES on Zinc.

We understand that many contents of the original manuscript seem more suitable to publish in next paper, once t-SMILES have been established. But as we pointed out before, we feel that it would be very difficult to convince the audience to accept our proposal to make any changes to the famous classical SMILES system, a de-facto standard for string-based representing molecular information in-silico. We hoped to try our best to provide evidence for our proposal.

Response 1.4 t-SMILES Vs. SMILES, DeepSMILES and SELFIES

We have conducted additional experiments on ChEMBL, ZINC, and QM9 and performed analyses focusing on the comparison of t-SMILES vs. SMILES, DeepSMILES, and SELFIES.

If you indeed improve on those three, or even just over SMILES and DeepSMILES, that would be interesting already.

Response 1.5 Encouragement

We appreciate the reviewer's kind feedback. Recently, pre-trained Transformer based language models (LMs) have demonstrated their ability to generate English text that closely resembles human writing.

For the English language, as an arbitrary combination of letters from the Latin alphabet will not necessarily lead to a valid word, in this sense, English is not robust, while for example, SELFIES is robust with respect to chemistry, as mentioned by the authors discussing the evolution from SMILES to SELFIES toward perfect robustness(Krenn et al., 2022).

We hope to borrow NLP models for solving chemical problems and believe that the original SMILES model was really the best starting point in selecting models for describing molecules. Besides making SMILES more robust with respect to chemistry as SELFIES does, we focus our efforts on encoding-decoding algorithms and enriching the chemical information embedded in the new molecule representation framework t-SMILES based on the fragmented molecule described with classical SMILES-type string.

Our experiments demonstrate that the t-SMILES family can outperform SMILES, DeepSMILES, and SELFIES in certain tasks that utilize appropriate singleton or hydride models and are well-optimized, particularly in low-resource datasets and goal-directed tasks. In contrast, M. Krenn et al. suggest that DeepSMILES and SELFIES require further optimization to match the performance of SMILES on some tasks due to their advanced grammar. Our systematic comparison experiments also support this claim.

Additionally, t-SMILES is an open framework that has the potential to integrate DeepSMILES and SELFIES if they prove to be advanced in some cases. The use of t-SMILES, in combination with an appropriate fragmentation algorithm, is expected to significantly aid chemists in their molecular modeling efforts.

Minor comments:

=====

- p5: why the validity of generated molecules is not 100%? Can't you shoot for this?

You are at **99%** already!

If you have a proper algorithm, I don't see why it would not work all the time.

Response 1.6 99%

Thank you for your comment. Although the t-SMILES code has improved performance by reducing long-term dependency grammar, there may still be cases of incorrect grammar in fragments because the generative model used in this study is based on probability. For this reason, we initially use >0.99 . However, the t-SMILES framework is technically capable of generating chemically valid molecules. Therefore, the validity score has been updated to 1.0.

- at this step, potential users will only be interested by an **open-source** encoder and decoder for t-SMILES; possibly in Python

Response 1.7 Open-source code

Thank you. Data used in this study and open-source code in python are available at:

<https://github.com/juanniwu/t-SMILES/>

Then original function names have been renamed as `encode_single()` and `decode_single()`.

- p7 and Fig 1: I don't even understand how you fragment molecules. Can you show several famous molecules fragmented / t-SMILES encoded by your approach.

caffeine, aspirin, paracetamol, etc.

From a single example, and if your explanations are not good enough, how are people supposed to follow what you are doing?

Humans, like deep-learning models, might need a few **examples** in order to generalize. The fragmentation in Fig 1 seems like heavy atoms were saturated with hydrogens after bonds were cut; some bonds were duplicated after breaking in two a fused ring system (what fragmentation algorithm on earth does this? and how to reconnect fragments after such a molecular "disintegration"?)

Response 1.8 Examples

Thank you for your comments.

We are sorry that the fragmentation procedure in Fig.1 seems difficult to understand, leading to the confusion that: "what fragmentation algorithm on earth does this? and how to reconnect fragments after such a molecular 'disintegration'?"

Actually, the fragmentation algorithms demonstrated in this work, including JTVAE, BRICS, MMPA, and Scaffold, were all published years ago. We simply use it and pay our attention to encoding-decoding algorithm.

We agree that the decomposition algorithm of JTVAE is somewhat complex and differs from BRICS, MMPA, and Scaffold in terms of chemical perspective. To make it easier to understand, Fig.1 has been revised using TSID algorithm. The original Fig.1 has been relocated to the Supporting Information file as SI.Fig.A.1.1 to demonstrate the procedure of TSSA with a very detailed explanation. MMPA instead of JTVAE is used as an example to decompose molecule in both figures. In addition, the molecule demonstrated in Fig.1 and SI.Fig.A.1.1 has also been revised to Celecoxib.

Furthermore, more examples, such as Aspirin, Caffeine, Paracetamol and a chiral molecule have been added to SI.A.1 to demonstrate how to decompose molecules using different fragmentation algorithms and t-SMILES codes. In addition, more comments on key points of different code algorithms have been added in revised manuscript.

Three coding algorithms are presented in the revised manuscript: The difference between TSSA and TSDY/TSID is that in TSSA, two fragments share one real atom as a connecting point, whereas TSDY and TSID use an attachment point (indicated by a dummy atom '*' with or without ID in the TS string) to illustrate how the group bonds. In fact, the dummy atom '*' with or without ID is shared by two pieces in TSDY and TSID.

TSDY and TSID use the same logic to generate t-SMILES strings and decode them back. However, since the representation of the fragment is a bit simpler, the encoding and decoding performance is a bit better than TSSA, as can be observed in Fig.1 and SI.A.4. In particular, TSID has been optimized to achieve a significantly low score for reconstruction novelty, making it easier to support open-ring, retrosynthesis and reaction prediction, etc.

SI.A.3 to SI.A.5 have been revised to provide a brief introduction to fragmentation algorithms. The paper focuses on a different topic, so we suggest interested readers refer to the original paper for a more detailed explanation of how to cut molecules into pieces.

- are there **constraints** on the molecular fragmentation scheme t-SMILES can use?
E.g. is **opening rings** OK?

Response 1.9 Opening-Rings

Thank you for bringing up this open-ring problem.

In theory, the t-SMILES framework has no limitations on molecular fragmentation schemes from an algorithmic point of view. In practice, we recommend performing a sanity test for any new fragmentation algorithm. It is unfortunate that none of the four publicly available fragmentation schemes (BRICE, MMPA, Scaffold, and JTVAE) used in this study support open-ring, which is the main reason why we did not claim to support open-ring.

Although TSSA could theoretically support open-ring, the logic is somewhat complex. We also realize that open-ring is an important chemical problem when studying retrosynthesis and reaction prediction problems, so we optimized t-SMILES with TSDY and TSID. TSDY and TSID use a dummy atom '*' with or without ID instead of the share atom to reduce complexity and achieve higher performance.

To evaluate the scalability and adaptability of t-SMILES for the open-ring problem, we use RBrics(Zhang et al., 2023) to fragment molecules and use TSID as the encoding-decoding algorithm. The experiments on ChEMBL are given in S.I.E.8 for reference. Fragmenting molecules is a complex problem, but not the main focus of this study. Therefore, the experiments are included in the Supplementary Information (SI) for reference and future research.

As mentioned above, the length of this paper is excessive. In response to this problem, TSDY and TSID need to be published. This change has resulted in some experiments being moved to the SI.

- p15: proper data augmentation for SMILES is via SMILES **randomization**; cf. Randomized SMILES strings improve the quality of molecular generative models. J Cheminform 11, 71 (2019).

<https://doi.org/10.1186/s13321-019-0393-0>

Is this what you call SMILES **enumeration**?

Response 1.10 SMILES enumeration

We thank the reviewer for pointing out this difference.

The SMILES randomization has the same meaning as the SMILES enumeration in our manuscript.

Reviewer #2 (Remarks to the Author):

In this paper, the authors introduce t-SMILES, a new molecular representation

that includes two additional special characters to the SMILES vocabulary to account for rings and branches in a way that does not require paired brackets or SELFIES tokenization.

t-SMILES uses breadth-first rather than the depth-first search used for constructing SMILES, to reduce the nesting depth of characters and reduce the need for generative models to capture long-range dependencies in the grammar.

They show benefits to generating valid and novel molecules when training generative models on this input representation.

Response 2.0 Key contributions

We deeply appreciate the reviewer's objective and concise summary of this study's key contribution.

Comments and questions

In line 259, it's stated that candidates are randomly selected when assembling pieces during reconstruction of a molecule from t-SMILES. What is the "recovery rate" for a common dataset (e.g. Zinc 250k) when converting back and forth from SMILES to t-SMILES? The inverse of this question may be answered in Table 2.

Response 2.1 Recovery rate

We thank the reviewer for this new metric.

- 1) If the definition of "recovery rate" is valid and unique molecules that are in the training dataset after reconstruction (using canonicalized SMILES), which serves as an inverse measure of "novelty" as specified in this study. In this scenario, to facilitate comparison, we use novelty for all experiments. Please refer Table 1 for novelty scores.

$$\text{novelty in this study} = \frac{\text{valid and unique molecules not in training set}}{10K \text{ molecules (including duplicated)}}$$

$$\text{recovery rate} = \frac{\text{valid and unique molecules in training set}}{10K \text{ molecules (including duplicated)}} = 1 - \text{novelty}$$

- 2) If the definition of "recovery rate" is whether a SMILES/t-SMILES pair is exactly recovered. Please refer S.I.E.1 for detailed result on ChEMBL.

$$\text{recovery rate} = \frac{\text{exactly same canonicalized SMILES}}{10K \text{ samples}}$$

$$\text{recovery rate} \approx 1 - \text{novelty}$$

The small error arises from two factors: 1) the samples were randomly selected, and 2) t-SMILES A can be reconstructed into SMILES B due to the presence of closely related molecules in training dataset.

SI.Table.E.1 Recovery rate of random reconstruction on 10K samples

Dataset	Frag Alg	Recovery Rate
ChEMBL	TSSA_J	0.122
	TSSA_B	0.307
	TSSA_M	0.114
	TSSA_S	0.104
	TSDY_B	0.781
	TSDY_M	0.275
	TSDY_S	0.541
	TSID_B	0.997
	TSID_M	0.997
	TSID_S	0.997

This table has been added as SI.Table.E.1. Additionally, the definition of Novelty in SI.B.3.2 has been revised for accuracy.

In line 288, are the authors saying that “&^” characters occur **frequently enough** in the tokenization for generative models to pick up on?

Response 2.2 Frequent occurrence of “&^” characters

We deeply appreciate the reviewer for bringing this matter to our attention.

We also recognize that this is really a tricky problem. If the frequent occurrence of “&^” characters is the key reason why generative models can pick them up correctly, then why can't generative models generate 100% correct '(' and ')' when they also receive the second highest scores in SMILES. Here we just want to point out that in t-SMILES the “&^” characters are playing a similar role when used as tokens as that of '(' and ')' in SMILES according to the frequency of occurrence. But the crux of the matter is not the frequency of occurrence rather than the design idea of t-SMILES framework itself, the latter was related to the challenge of the syntactic invalidity associated with unbalanced parentheses. The newly introduced symbols '&^' in t-SMILES do not need to be paired, while '(' and ')' require pairing, causing SMILES syntax to have deep recursion. t-SMILES using smaller SMILES fragment reduces the long-term dependencies in the grammar, and simplifies the overall complexity. So, we also calculated Nesting Depth.

This is a very basic understanding. However, proving it in theory goes beyond the scope of this study and could be a target for future research. Although t-SMILES and classical SMILES appear similar, there are some fundamental differences between them.

On the other hand, the addition of "&^" symbols does not present a new challenging issue, like the scarcity reasoning problem, as evidenced by their high frequency.

Analyzing deep learning models is a complex and arduous topic. Regrettably, we needed to exclude some basic discussion, such as section 3.1.2 on Entropy, due to length limitations.

Nevertheless, we are continuing our research to improve our understanding of the fundamental properties of t-SMILES from both an algorithmic and chemical perspective.

In response to this issue, we have revised it as:

“It means that, a crucial task of the SMILES based model is to learn and predict paired '(' and ')' symbols. Conversely, the t-SMILES based model must learn how to reason non-paired symbols '&' and '^'.”

Because the authors refer to t-SMILES in the tables using the **names** of the approaches they use for substructure identification (e.g., JTVAE), I’d suggest highlighting those columns and somehow indicating that they are all flavors of t-SMILES.

Response 2.3 Abbreviations

We are most grateful for your kind suggestion. In the revised manuscript, some new names and abbreviations have been used to describe t-SMILES algorithms, such as TSSA_J means TSSA-style code algorithm that uses JTVAE as a fragmentation algorithm.

In line 325, what is meant by “models that do not require training...introduce other problems such as long training time” ?

Response 2.4 Training molecules

We thank the reviewer for bringing our attention to the omission of the keyword "training molecules". This has been corrected as following:

“The authors point out: models that do not require training molecules are free from this problem,...”

How does t-SMILES compare to **CREM** and **Group SELFIES**? These approaches also introduce chemical diversity through either “chemically reasonable mutations” (CREM) or assembling fragments (Group SELFIES).

We thank the reviewer for suggesting these analyses. These two articles have been cited. We discuss them separately in two items: Response 2.5 and Response 2.6.

Response 2.5 CReM

CReM presents an intriguing approach to generating molecules from fragment data. However, the authors note that “which less suitable for a stochastic sampling of compounds similar to ones used for generation of a fragment database, and other major limitation of the current implementation is the inability to create new ring systems so the performance depends on their representativeness in the input compound database”.

1. Rigorous and systematic experiments on goal-directed learning have been added in revised manuscript, where CReM serves as a key baseline for evaluating t-SMILES algorithms.
2. Experiments on t-SMILES using goal-directed reconstruction exhibit the potential for achieving better scores than CReM through appropriate optimization processes. More comprehensive experiments and nuanced discussions, tables and figures have been added, like Table4 & Fig.4.
3. In fact, the CReM approach differs from t-SMILES in that it seems to struggle with learning the distribution of the training data. On the contrary, t-SMILES could effectively learn the distribution of the training data.
4. t-SMILES has the ability to generate fragments, including new rings, that can serve as a foundation for CReM. It would be an interesting solution in future research to use a simple process of CReM to reconstruct fragments in the t-SMILES process. CReM and t-SMILES framework would complement each other to get better performance.

Table 4. Results of Goal-Directed Benchmarks on ChEMBL.

ID	BN	DSs	CReM	SMG	DSG	SFG	TSBR	TSBG	TSSR	TSSG	TSMR	TSMG
9	MM1	0.297	0.360	0.453	0.383	0.410	0.446	0.421	0.443	0.452	0.438	0.467
16	SMPO	0.496	0.708	0.598	0.625	0.725	0.692	0.755	0.693	0.788	0.866	0.930
18	VS	0.258	0.987	0.985	0.988	0.986	0.991	0.987	0.994	0.997	0.985	0.996

Fig.4 Performance of the Goal-Directed Benchmarks for T16.SMPO with different training epochs. The TSMG model yields significantly high results. For further comprehensive experiments and nuanced discussions, please refer to section SI.B.3.4.

Response 2.6 Group SELFIES

Group SELFIES is a dictionary ID based solution that relies on SELFIES and fragments. While some research shows that SELFIES outperform SMILES, other studies(Krenn et al., 2022)(Chen

et al., 2023) suggest that besides validity, other metrics are difficult to be optimized to outperform classical SMILES, and its advanced grammar makes some strings are difficult to parse.

Moreover, it is evident that models based on dictionary IDs suffer from some fundamental problems, such as in-vocabulary (IV), out-of-vocabulary (OOV), and high-dimensional sparse representation (curse of dimensionality).

In response we have revised the Introduction to include a reference and highlight the common problem with solutions relying on dictionary IDs.

In addition, we have included it as a baseline mode for comparison with the t-SMILES model in experiments on Zinc. It reads: “As to Group-VAE, which uses fragments and SELFIES, it achieves a lower novelty score compared to SELFIES-VAE in published experiments. Because Group-VAE uses a different way to calculate FCD, it is impossible to make a direct comparison. But in this experiment, SELFIES based model obtains lower scores for both novelty and FCD when compared to five t-SMILES based models.”

In Fig 6 you might plot **Novelty against FCD** score and use **colors and shapes** to indicate methods and training epoch, to better show the tradeoff.

Response 2.7 Figure of Novelty-FCD

We are most grateful for your suggestion. Figure has been revised as below.

The **t-SNE** plots in Fig 7 do not have any clearly identifiable clustering structure and may be better suited to the **SI** if the conclusion is just that training and generated samples overlap.

Response 2.8 Figure of JNK3

We are most grateful for your suggestion. Fig 7 has been relocated to SI.D.1.

Reference

- Cadeddu, A., Wylie, E. K., Jurczak, J., Wampler-Doty, M., & Grzybowski, B. A. (2014). Organic chemistry as a language and the implications of chemical linguistics for structural and retrosynthetic analyses. *Angewandte Chemie International Edition*, 53(31), 8108–8112.
- Chen, Y., Wang, Z., Zeng, X., Li, Y., Li, P., Ye, X., & Sakurai, T. (2023). Molecular language models: RNNs or transformer? *Briefings in Functional Genomics*, 22(4), 392–400.
<https://doi.org/10.1093/bfgp/elad012>
- Jean-Marie Lehn - Interview*. (n.d.). Retrieved January 8, 2024, from
<https://www.nobelprize.org/prizes/chemistry/1987/lehn/interview/>
- Krenn, M., Ai, Q., Barthel, S., Carson, N., Frei, A., Frey, N. C., Friederich, P., Gaudin, T., Gayle, A. A., Jablonka, K. M., Lameiro, R. F., Lemm, D., Lo, A., Moosavi, S. M., Nápoles-Duarte, J. M., Nigam, A. K., Pollice, R., Rajan, K., Schatzschneider, U., ... Aspuru-Guzik, A. (2022). SELFIES and the future of molecular string representations. *Patterns*, 3(10). <https://doi.org/10.1016/j.patter.2022.100588>
- Lehn, J.-M. (1988). Supramolecular chemistry — Scope and perspectives: Molecules — Supermolecules — Molecular devices. *Journal of Inclusion Phenomena*, 6(4), 351–396.
<https://doi.org/10.1007/BF00658981>
- Zhang, L., Rao, V., & Cornell, W. (2023). r-BRICS – a revised BRICS module that breaks ring structures and carbon chains. *ChemMedChem*, e202300202. <https://doi.org/10.1002/cmdc.202300202>

REVIEWER COMMENTS

Reviewer #1 (Remarks to the Author):

Major remarks:

=====

The authors present many different versions of their method (TSSA-*). Why don't you just show a maximum of two recommended version: e.g. one for optimization tasks (goal-oriented benchmarks), the other for distribution reproduction experiments. Readers will probably not care about all the other variants you are proposing. I guess they were explored as part of the research; but readers are mostly interested by the final result. Maybe you also need to recommend a molecular fragmentation scheme with each by the way, since your method is parameterized by such a scheme.

Minor remarks:

=====

- maybe better title: "t-SMILES: a fragment-based molecular representation framework for de novo ligand design"
 - abstract: outmaneuver overfitting problem -> avoid overfitting
 - abstract: pre-training fine-tuned -> pre-trained then fine-tuned
 - p3: outmaneuver -> avoid the
 - Fig 4: replace by a table; there are too many curves, I don't see an obvious trend
 - figures on p16 and p17: I don't understand the difference. Fig from p16 should probably be the main figure of the paper.
 - figure p20: use a table instead; too many curves; same as before
 - in the supplementary data: if you show 100 molecules generated from ChEMBL, then also show just before 100 molecules from ChEMBL, so that we can see the similarity. Same for molecules generated from some other chemical datasets.
 - I have no idea about the training time of their model (how long to train per epoch, for how many molecules in the training set and on what kind of computer)
 - I have no idea about the molecular generation time of their model (in molecule/s); also, do they need/benefit from executing on a GPU.
 - I know at least two prior fragment-based molecular generator which are not cited:
 - * Break Down in Order To Build Up: Decomposing Small Molecules for Fragment-Based Drug Design with eMolFrag
<https://pubs.acs.org/doi/10.1021/acs.jcim.6b00596>
 - * Molecular generation by Fast Assembly of (Deep)SMILES fragments
<https://jcheminf.biomedcentral.com/articles/10.1186/s13321-021-00566-4>
- The authors cite CReM [40] and PMLR 2020 [26], but that's an incomplete state of the art.

Reviewer #2 (Remarks to the Author):

I appreciate the authors' considered responses to my comments. All my comments have been addressed.

Dear Reviewers,

We thank reviewers for their valuable and constructive comments on the revision of our manuscript entitled "t-SMILES: A Scalable Fragment-based Molecular Representation Framework for De Novo Molecule Generation "(NCOMMS-22-30046B). The manuscript has been renamed to "t-SMILES: A Fragment-based Molecular Representation Framework for De Novo **Ligand Design**".

We have addressed all concerns that arose during the review process and provided a detailed point-by-point response to these comments and suggestions. We list the major changes below:

1) In response to the major comments, we conducted rigorous experiments, nuanced discussions, and presented the best performing models:

TSSA_S for goal-oriented tasks, TSID with a hybrid fragmentation scheme for distribution reproduction experiments, and TSDY_M for balanced goal-oriented and distribution reproduction tasks.

2) As to the two additional references:

The first one, eMolFrag, is a molecular fragmentation tool based on BRICS algorithm, which has been cited in revised manuscript.

The second one, FASMIFRA, generates molecules using a fragment assembly method. It has been cited and compared with t-SMILES models in revised manuscript.

In addition, another model, eSynth, which is used in the eMolFrag paper to assemble fragments, has been also cited because its statements fit perfectly with our goal of performing experiments on low-resource datasets.

REVIEWER COMMENTS

Reviewer #1 (Remarks to the Author):

Major remarks:

=====

The authors present many different versions of their method (TSSA-*).

Why don't you just **show a maximum of two recommended version**: e.g. one for optimization tasks (**goal-oriented benchmarks**), the other for **distribution reproduction** experiments.

Readers will probably not care about all the other variants you are proposing.

I guess they were explored as part of the research; but readers are mostly interested by

the final result.

Maybe you also need to **recommend a molecular fragmentation scheme** with each by the way, since your method is parameterized by such a scheme.

Response 1.0 two recommended TS* version

We appreciate the reviewer's thoughtful review and constructive proposals, which helped us thoroughly and comprehensively examine and summarize the t-SMILES system.

We acknowledge that readers are primarily interested in the final outcome and concur with the reviewer's proposal to classify the t-SMILES algorithm into two categories. The experiments in main text are actually a proposal for various t-SMILES algorithms and fragmentation schemes. We apologize for not point this out more clearly.

The three code algorithms are evaluated using distribution learning metrics to gain a basic understanding. And then, TSSA_S is selected for low-resource tasks, TDSY is used for general goal-oriented benchmarks, and TSID is specifically adopted for the open-ring problem with an additional fragmentation scheme.

If for general purpose, the proposal for two types of tasks would be:

1. For goal-oriented tasks, the preferred option is **TSSA_S** to avoid 'striking similarity' to the training dataset and achieve 'better novelty with reasonable similarity'.
2. Different TSDY models yield balanced scores, so it would be better to choose algorithm based on chemical purpose. If to be simple, **TSDY_M** is proposed as the optimal choice for balanced goal-oriented and distribution reproduction tasks.
3. For distributional reproduction tasks, it is preferred to use **TSID with a hybrid fragmentation scheme** to explore chemical spaces as much as possible and achieve high similarity to the training dataset.

In real-world molecular modeling tasks, chemists often face challenging problems with far more potential choices than can be explored in a systematic way. The t-SMILES framework allows for the design of models using multi-code combination strategies or multi-model systems with different fragmentation schemes. As demonstrated in 'Chemical Space' and 'Goal-Directed Learning', t-SMILES-based models can comprehensively explore the complexity and diversity of chemical space. In addition to the aforementioned proposal, users can also design their own t-SMILES models based on the empirical results of this study to achieve specific experimental goals.

In order to cautiously respond to this suggestion, the following actions have been taken in the revised manuscript.

1. Some additional experiments on TSSA, TSDY and TSID have been conducted in the revised manuscript, including:

a. TSDY on the low-resources dataset JNK3 (SI.Table.D.4.2 on page 48 of SI).

The results indicate that TSDY-based models receive significantly lower Active-Novel scores compared to TSSA-based models, with a score of 0.275 for TSDY versus 0.565 for TSSA.

b. TSSA on goal-oriented benchmarks on ChEMBL (SI.B.3.3.5 on page 31-32 of SI).

The results indicate that TSSA-based models, similar to TSDY-based models, can outperform models based on SMILES, DeepSMILES, SELFIES and baseline model CREM.

c. TSID on goal-oriented benchmarks on ChEMBL (SI.B.3.3.5 on page 31-32 of SI).

The results indicate that TSID-based models have limited ability compared to TSSA and TSDY based models for goal-oriented tasks.

2. The previous statement in **Discussion** section has been revised to clearly point out the proposal with a new statement, as shown below marked in green:

Three code algorithms are presented in this work: TSSA, TSDY, and TSID. TSSA is recommended as the first choice for goal-directed tasks, especially for one-round low-resource tasks, to avoid 'striking similarity' to the training dataset and achieve 'better novelty with reasonable similarity'. TSID is recommended for distribution reproduction experiments due to its accurate fit to the physicochemical properties of the training data. TSDY codes are a balanced choice for both goal-oriented and distributional reproduction tasks.

3. As the experiments of TSDY have been included in the revised manuscript, the following statement has been removed from the second to last paragraph of the **Experiments on Low-Resource Datasets** section in the main text:

~~Furthermore, it is important to note that TSDY can better fit the physicochemical properties of the training data compared to TSSA. Therefore, it can be considered for future research.~~

4. The proposal has been added to the SI file and can be found in the summary on pages 14-15.

To ensure cautious and objectivity, we would like to respond to this comment in detail from two angles, including chemical and algorithms. The comprehensive explanation is a little long, we are sorry about it.

Molecular fragmentation schemes are based on specific chemical principles. From this perspective, it is not easy to make a definitive judgment about which option is superior to another. Furthermore, in real-world molecular design experiments, the goal is often to address a specific problem, such as designing a molecule with a particular scaffold. Similarly, chemists may wish to perform a thorough investigation of open ring problems. In these scenarios, different fragmentation schemes should be selected to accomplish the given task. For example, if the target task is related to open rings, none of BRICS, MMPA, Scaffold, or JTVAE are suitable, only algorithms such as RBrics, which can cut rings, are the best choice.

Although it was only a **minor** comment (concerning the open-ring problem) in the last revision, we took it seriously and published TSDY and TSID. Our action, however, actually contradicts your **major** suggestion to ‘dump all the rest, or try to publish the rest in a future paper, once t-SMILES have been established’. We took this risk really after careful consideration. Our belief is that this action benefits readers by providing a comprehensive overview of the t-SMILES framework, although much more experiments are required. We appreciate the reviewer providing additional ideas from a chemical perspective, which has made the t-SMILES code system more flexible and versatile.

From systemically experiments and nuanced discussions, it is evident that TSSA, TSDY, and TSID have different distinct advantages at different points. They complement each other well but cannot be completely replaced by one another. A figure with highlighted features is included for a visual aid in the discussion.

Fig.R.1

Detailed experiments conducted on ChEMBL, Zinc, and QM9 show that TSID models achieve similar scores to TSDY models for distribution learning tasks. They both outperform TSSA models due to their simpler structure. And the results of the additional experiments during this revision process demonstrate that TSSA-based models, similar to TSDY-based models, can outperform baseline models in goal-oriented

benchmarks as well. However, from the additional experiment we can see that the TSID-based models have limited ability compared to TSSA and TSDY based models for goal-oriented tasks.

SI.Table.B.3.3.5 Results of the Goal-Directed Benchmarks for T16.SMPO with different training epochs.

Round	SMG	DSG	SFG	TSBR	TSBG	TSSR	TSSG	TSMR	TSMG	TSSA_BG	TSSA_MG	TSSA_SG	TSSA_MR	TSSA_MR	TSID_M
1	0.505	0.505	0.505	0.505	0.505	0.505	0.505	0.505	0.505	0.505	0.505	0.505	0.505	0.505	0.505
2	0.505	0.505	0.507	0.505	0.505	0.505	0.505	0.506	0.505	0.505	0.505	0.509	0.505	0.505	0.505
3	0.505	0.505	0.507	0.505	0.515	0.505	0.506	0.506	0.505	0.505	0.505	0.514	0.505	0.508	0.505
4	0.506	0.506	0.508	0.505	0.515	0.506	0.512	0.506	0.505	0.505	0.505	0.516	0.505	0.508	0.505
5	0.506	0.506	0.509	0.505	0.515	0.507	0.515	0.506	0.505	0.508	0.507	0.523	0.507	0.509	0.505
10	0.521	0.512	0.519	0.516	0.518	0.528	0.538	0.510	0.505	0.518	0.516	0.575	0.515	0.538	0.505
20	0.544	0.529	0.560	0.543	0.586	0.566	0.600	0.548	0.523	0.540	0.547	0.656	0.547	0.618	0.505
30	0.555	0.552	0.576	0.589	0.611	0.602	0.637	0.574	0.556	0.564	0.590	0.687	0.562	0.643	0.505
45	0.561	0.586	0.632	0.610	0.699	0.644	0.681	0.620	0.621	0.605	0.774	0.770	0.600	0.676	0.505
50	0.565	0.590	0.646	0.612	0.716	0.656	0.701	0.639	0.717	0.608	0.812	0.778	0.620	0.685	0.505
90	0.596	0.624	0.723	0.688	0.745	0.689	0.780	0.866	0.928	0.659	0.816	0.780	0.713	0.756	0.506
100	0.598	0.625	0.725	0.692	0.755	0.693	0.788	0.866	0.928	0.669	0.830	0.780	0.717	0.761	0.506

Considering your suggestion to show a maximum of two recommended versions, it may be possible to remove TSID entirely. However, doing so could make it more challenging to solve the open-ring problem. Therefore, we suggest keeping three algorithms in this study.

Based on the comparative analysis, it is clear that the TSSA code is the first choice for goal-directed tasks, especially for one-round low-resource tasks. The TSSA algorithm has some remarkable advantages that cannot be replaced by TSDY. While TSID models achieve better FCD scores than TSSA models, it's recommended as the first choice in distribution reproduction experiments. The TSDY code is a balanced choice for both goal-oriented and distributional reproduction tasks.

We hope these additional experiments and nuanced discussions will help alleviate any concerns and apologize for any confusion caused by the numerous experiment results obtained using different algorithms.

Minor remarks:

=====

- maybe better title: "t-SMILES: a fragment-based molecular representation framework for de novo **ligand design**"

Response 1.1 title

We are most grateful for reviewer's thoughtful and constructive suggestion.

In the revised manuscript, title has been updated.

-
- abstract: outmaneuver overfitting problem -> avoid overfitting
 - abstract: pre-training fine-tuned -> pre-trained then fine-tuned
 - p3: outmaneuver -> avoid the

Response 1.2 words correction

We are most grateful for reviewer's kind and professional suggestion. All of them have been updated in the revised manuscript.

-
- Fig 4: replace by a table; there are too many curves, I don't see an obvious trend

Response 1.3 Fig.4 table and curves

We would like to apologize for any confusion caused by the abundance of models. Fig.4 visualizes a kernel density estimation of these distributions, similar to the figures used in the MOSES benchmark.

To enable easy comparison, these three figures utilize the same baseline model as the first one. This baseline model consists of 10,000 randomly selected molecules from the training dataset. The Wasserstein distance for this baseline model is set as zero. Tiny system errors are a result of random sampling. Same goes for Zinc.

To quantitatively compare the distributions in the generated and test sets, we compute a 1D Wasserstein distance between property distributions of generated and test sets. The tables that correspond to Fig. 4 were previously located in the SI file under SI.Table.E.4 for ChEMBL and SI.Table.E.5 for Zinc.

SI Table.E.4 Wasserstein distance metrics for baseline description, baseline models and t-SMILES family. The worst (higher) scores for each task are highlighted in bold red, scores that are better (lower) than classic SMILES are highlighted in bold blue.

Model	MolWt	LogP	pLogP	SAS	NPS	QED	BertzCT	TPSA	FCSP3
SMILES	3.514	0.065	0.074	0.006	0.012	0.006	8.897	1.218	0.009
DSMILES	10.734	0.127	0.119	0.023	0.017	0.020	43.891	1.088	0.003
SELFIES	5.065	0.047	0.296	0.032	0.019	0.007	43.275	1.569	0.009
hg2G	76.592	0.583	0.363	0.114	0.022	0.088	266.483	16.723	0.016
TSSA_J	5.175	0.264	0.691	0.116	0.070	0.018	53.273	1.233	0.026
TSSA_B	10.701	0.213	0.376	0.101	0.072	0.040	38.879	4.852	0.007
TSSA_M	13.034	0.230	0.469	0.061	0.059	0.007	31.481	2.545	0.008
TSSA_S	5.273	0.145	0.425	0.071	0.041	0.014	9.710	4.508	0.020
TSDY_B	8.366	0.078	0.119	0.010	0.015	0.021	11.378	2.846	0.007
TSDY_M	13.236	0.114	0.088	0.017	0.026	0.011	21.356	3.422	0.004
TSDY_S	5.929	0.075	0.118	0.007	0.011	0.004	9.026	1.775	0.017
TSID_B	5.096	0.038	0.113	0.010	0.010	0.005	13.450	1.594	0.003
TSID_M	16.193	0.084	0.092	0.022	0.007	0.020	48.175	4.705	0.006
TSID_S	3.352	0.045	0.075	0.003	0.005	0.009	6.311	1.052	0.004
TS_Vanilla	4.859	0.095	0.105	0.005	0.012	0.007	20.737	3.167	0.017
TSSA_H5	7.632	0.123	0.349	0.065	0.044	0.008	21.625	1.513	0.008
TSDY_H4	7.400	0.199	0.245	0.010	0.016	0.012	13.536	2.481	0.007
TSID_H4	5.966	0.076	0.141	0.016	0.006	0.010	10.239	1.766	0.006
TSDY_HBV	6.739	0.159	0.182	0.004	0.011	0.016	14.613	0.807	0.004
TSDY_HMV	7.552	0.052	0.091	0.009	0.015	0.009	10.308	3.209	0.011
TSDY_HSV	4.984	0.196	0.268	0.011	0.011	0.006	20.962	2.920	0.023
TSID_HBV	5.190	0.050	0.096	0.007	0.016	0.004	9.566	2.320	0.006

SI Table.E.5 Wasserstein distance metrics for baseline description, baseline models and t-SMILES family. The worst (higher) scores for each task are highlighted in bold red, scores that are better (smaller) than classic SMILES are highlighted in bold blue.

Model	MolWt	LogP	pLogP	SAS	NPS	QED	BertzCT	TPSA	FCSP3
SMILES	4.155	0.056	0.049	0.004	0.009	0.006	8.820	1.764	0.011
DSMILES	1.392	0.048	0.065	0.004	0.010	0.003	7.221	0.741	0.010
SELFIES	2.926	0.042	0.079	0.026	0.008	0.009	5.392	0.886	0.005
JTVAE	48.111	0.819	1.144	0.098	0.112	0.023	137.136	2.624	0.034
FragDgm	62.137	0.529	0.914	0.168	0.087	0.148	153.386	13.670	0.063
BRICS_Rnd	10.956	0.319	0.588	0.123	0.102	0.022	37.722	3.728	0.040
TSSA_J	4.051	0.159	0.390	0.067	0.069	0.002	19.702	2.139	0.009
TSSA_B	5.315	0.124	0.297	0.081	0.086	0.046	15.353	4.212	0.005
TSSA_M	3.237	0.195	0.407	0.064	0.077	0.026	6.216	4.426	0.005
TSSA_S	3.918	0.203	0.486	0.083	0.069	0.038	13.763	6.974	0.003
TSDY_B	4.222	0.058	0.067	0.010	0.013	0.013	8.778	0.694	0.007
TSDY_M	6.311	0.121	0.096	0.015	0.028	0.014	24.816	1.636	0.013
TSDY_S	2.308	0.040	0.045	0.007	0.010	0.010	9.833	0.686	0.010
TSID_B	5.648	0.043	0.038	0.010	0.017	0.021	18.274	0.817	0.006
TS_Vanilla	3.892	0.032	0.062	0.006	0.012	0.003	8.992	1.183	0.005
TSSA_H5	4.430	0.138	0.340	0.052	0.062	0.022	11.391	2.838	0.004
TSDY_H4	6.796	0.065	0.040	0.018	0.016	0.016	19.598	0.872	0.005
TSDY_HBV	2.699	0.073	0.086	0.003	0.003	0.010	10.297	0.274	0.004
TSDY_HMV	3.711	0.072	0.061	0.010	0.012	0.008	15.373	0.520	0.010
TSDY_HSV	1.703	0.031	0.045	0.005	0.006	0.007	7.532	0.444	0.011
TSID_HBV	4.550	0.030	0.044	0.008	0.004	0.007	9.618	0.358	0.005

We acknowledge that the Fig.4 may be difficult to read. Since the main text permits a maximum of 10 figures and tables, we have selected only SAScore and merged the curves of ChEMBL and Zinc into one

figure. If tables are preferred, we need to combine SI.Table.E.4 for ChEMBL and SI.Table.E.5 for Zinc. These tables are quite large, with over 40 rows.

In addition, the Wasserstein distance is a single value with limited information, whereas Figure 4 presents a range that could provide more comprehensive information.

We hope this clarification will help alleviate any concerns. In response to this issue, Fig. 4 has been revised for clarity by removing some curves.

- 1) The curves of the hybrid models represent an average of various algorithms, which are demonstrated in C3 and Z3.
- 2) Only one algorithm of different t-SMILES algorithms is demonstrated in C2 and Z2.

The updated figure is presented below.

- figures on p16 and p17: I don't understand the difference.

Fig from p16 should probably be the **main figure** of the paper.

Response 1.4 main figure

The figure on page 16 is the full-size image of Figure 1. for TSDY and TSID. This is the main figure in the main text.

The figure on page 17 is the full-size image of SI.Fig.A.1.1. for the TSSA, which is located in SI.

We apologize for the confusion. In the revised manuscript, we have added the figure ID on full size image and removed the full-size image of SI.Fig.A.1.1.

- figure p20: use a table instead; too many curves; same as before

Response 1.5 Fig p20

Figure p20 is the full-size image of Fig.4.

We apologize for any confusion. In the revised manuscript, we have added the Figure ID on full size image.

- in the supplementary data: if you show 100 molecules generated from ChEMBL, then also show just **before 100 molecules from ChEMBL**, so that we can see the similarity. Same for molecules generated from some other chemical datasets.

Response 1.6 molecules from ChEMBL

Randomly selected molecules from the training dataset have been added to the revised SI file as F.1.0, F.2.0 and F.3.0.

We would like to mention that the molecules in Figures are randomly selected from both the training dataset and the generated group. Therefore, the selected molecules may differ slightly from those displayed in the previous manuscript.

-
- I have no idea about the **training time** of their model (how long to train per epoch, for **how many molecules in the training set** and on **what kind of computer**)
- I have no idea about the molecular **generation time** of their model (in molecule/s); also, do they **need/benefit from executing on a GPU**.

We thank the reviewer for the meticulous review and comments. We will response to these comments in the different items below.

Response 1.7 how many molecules in the training set

Section B.2 of the SI file had included detailed information on the datasets, including the number of molecules in the training set, which is listed in the second column of Table B.2.1.

Dataset	Number of Samples	N_Characters		N_Atoms		N_Rings	
		Max	Mean	Max	Mean	Max	Mean
Zinc	249454	120	44.4	38	23.2	9	2.8
QM9	133017	37	16	9	8.8	8	1.7
ChEMBL	1570407	120	48	111	28	30	3.4

Response 1.8 training and generation time

Since t-SMILES strings are similar to SMILES strings, training time is almost similar. Therefore, we did not include training and generation time in the original manuscript.

In response to this comment, more information as follows has been added in revision SI file section B.1 (page 18-19).

We recognize that both of the recommended reference papers eSynth[2] and FASMIFRA[1] emphasize the time required to generate one molecule. Both of them use faster programming languages to get better performance.

1. eMolFrag & eSynth (the first) use the programming language C++, which is faster than Python.
2. FASMIFRA (the second) uses the programming language OCaml to run faster (than python) indexing of fragments and perform fragment assembly (p4).

In our source code, the reconstruction of t-SMILES to generate molecules is separated from the rest of the process as a standalone task. This task can be optimized later using faster programming languages and advanced techniques. We thank the reviewer for carefully examining our manuscript and pointing out this issue. We could consider investigating it in-depth and publishing it in a future paper once t-SMILES have been established.

Because the principles for generating molecules of t-SMILES and the recommended papers are different, we calculate the cost time as two separate parts to make it easier to compare.

The corresponding revision is as follows:

Due to the similarity between t-SMILES and SMILES strings, training time is not a key indicator to evaluate their differences. For instance on GPU: NVIDIA Quadro RTX4000, CPU: Intel(R) Xeon(R) W-2265, a classical SMILES-based model takes almost 24 hours to train, while the TSID_S model takes almost 19 hours with the same training parameters: epochs = 10, batch_size = 128, and tokens such as B, Br, C, Cl, F, I, N, O, P, S, [Cl+2], [Cl+3], etc.

The cost of training depends on various factors such as hyperparameters and tokenization method. It is important to note that the reference run time is not an indication that the t-SMILES model requires less time than the SMILES model, but rather serves as a reference point.

When generating 1000 molecules on an GPU: NVIDIA Quadro RTX4000 with a batch size of 128, the TSID_S model takes approximately 40 seconds, resulting in roughly 25 molecules per second. While, the SMILES model takes around 20 seconds and produces approximately 50 molecules per second. The difference is mainly because the TSID_S and SMILES models have different token dictionaries.

One key difference between t-SMILES and SMILES is that the t-SMILES model must reconstruct the string into molecules, which distinguishes it from the SMILES-based model. The cost of reconstructing 1000 molecules is shown in SI.Table.B.1.2.3.

SI.Table.B.1.2.3 Cost to reconstruct molecules.

Code	TSSA_J	TSSA_B	TSSA_M	TSSA_S	TSDY_B	TSDY_M	TSDY_S	TSID_B	TSID_M	TSID_S
s/1000Mols Candidate=1	74	22	48	172	18	47	24	17	43	22
s/1000Mols Candidate=3	133	31	150	356	27	120	43	17	43	21
Molecule/s Candidate=1	14	45	21	6	56	21	42	59	23	45
Molecule/s Candidate=3	8	32	7	3	37	8	23	59	23	48

*The term 'candidate' refers to the number of sub-fragments selected for the next step during reconstruction.

Response 1.9 need/benefit from executing on a GPU

In SI file, we included information about training time and GPU usage in the last revision. The details are as follows:

The t-SMILES model TSID_S_[R10] achieves the highest FCD score of 0.891 among t-SMILES models. We evaluate its repeatability by running the model three times (3*24 hours on NVIDIA Quadro RTX4000) on different GPUs. A summary of metrics can be found in SI.Tabel.B.1.2.2.

SI.Fig.B.3.3.4.2 Performance of the Goal-Directed Benchmarks for T16.SMPO with different training epochs. The TSMG yields significantly high results, and therefore, we conduct multiple executions to confirm it thoroughly. S: GPU 1: NVIDIA GeForce RTX 3090, D: GPU 2: NVIDIA Quadro RTX4000

In response to this comment, more information as follows has been added in revision SI file section B.1 (page 18). It reads:

In this study, two GPUs: NVIDIA GeForce RTX 3090 and NVIDIA Quadro RTX4000 are used to train deep learning models. The encoding and decoding of t-SMILES string can be performed on a CPU without the need for a GPU. To improve performance, future studies could explore the use of parallel techniques or other programming languages, such as C++, which are faster than Python.

- I know at least two prior fragment-based molecular generator which are not cited:

* Break Down in Order To Build Up: Decomposing Small Molecules for Fragment-Based Drug Design with **eMolFrag**

<https://pubs.acs.org/doi/10.1021/acs.jcim.6b00596>

* Molecular generation by Fast Assembly of (Deep)SMILES fragments

<https://jcheminf.biomedcentral.com/articles/10.1186/s13321-021-00566-4>

The authors cite CReM [40] and PMLR 2020 [26], but that's an incomplete state of the art.

We thank the reviewer for bringing these to our attention and apologize for the missing.

Response 1.10 eMolFrag

1. eMolFrag is a molecular fragmentation tool based on BRICS algorithm written in Python. So, in the revised manuscript, the SI file cites the eMolFrag as a fragmentation algorithm (page 12).

The newly added statement is shown below, marked in green:

The study of fragmentation methodologies and their applications continues to reveal new opportunities for efficient molecular design and development. A review paper summarized a total of 15 published algorithms, such as eMolFrag etc.

2. In addition, another fragment assembling solution eSynth[2], which comes from this recommendation, mentioned that:

This protocol mimics a real application, where one expects to discover novel compounds based on a small set of already developed bioactives.

And some statements in Background:

These large generic collections, like Zinc, have a very low probability to exhibit the desired bioactivity for a specific target protein. Consequently, the chances to identify novel, high-quality leads from large compound repositories are low.

These statements align perfectly with our goal of conducting experiments on low-resource datasets. Our experiments on JNK3 also support these statements. It is evident that the SMILES-based pre-trained then fine-tuned models are not the best ones to achieve higher active-novelty scores. In the revised manuscript, eSynth has also been cited as a key reference. We would like to express our sincere appreciation to reviewer's constructive suggestion and bring this valuable reference to us.

The following statement has been added to the first paragraph of **Experiments on Low-Resource Datasets** and is marked in green:

The scarcity of labeled data presents a challenge for implementing deep learning in target-oriented drug discovery. This section simulates a real-world scenario, where novel compounds are discovered based on a small set of pre-existing bioactive compounds. This is because in the generic large compound libraries, such as Zinc, the vast majority of compounds have a very low probability to exhibit the desired bioactivity for a specific target protein. As a result, the chances to identify novel, high-quality leads from large compound repositories are low.

Response 1.11 FASMIFRA

This is a valuable solution to assemble fragments as a valid molecule, particularly with the concept of 'extended bond typing'. However, the method described in the paper **does not generate new rings**, as noted by the author. Meanwhile it doesn't create new fragments as well.

FASMIFRA[1] proposes a different solution to the deep generative model used in this study. By combining the benefits of t-SMILES, it is possible to achieve improved results, which could serve as a starting point for further research.

This solution has been cited as a baseline model in distribution learning table (**Table 3**) in newly revised manuscript. In addition, the following statement has been added to the second paragraph of **Distribution Learning on ChEMBL** and is marked in green:

When comparing t-SMILES models with fragment-based assembling algorithm: FASMIFRA, all TSDY and TSID based models outperform it in both novelty and FCD dimensions. However, this kind of assembly algorithm is expected to be used as a reconstruction process for the t-SMILES framework to improve performance in the future.

Response 1.12 CReM and PMLR 2020(FragDgm)

We appreciate the reviewer's careful examination of our manuscript and their feedback on this issue.

1. PMLR 2020 [26] (FragDgm) encodes the dictionary-ID of fragments as a string. The manuscript has included the drawbacks of this kind of solution that uses dictionary-ID, the 6th paragraph in Introduction. It reads: *It is obvious that dictionary-ID-based models suffer from some fundamental problems such as in-vocabulary (IV), out-of-vocabulary (OOV), and high-dimensional sparse representation (curse of dimensionality).*

While, PMLR 2020 (FragDgm) uses language model, we listed it as a baseline model in the original manuscript as we use language model in this study to evaluate t-SMILES as well.

In response to this concern, in revised manuscript:

- a) the curve of FragDgm has been removed from Fig.4.
- b) the comments have been removed from manuscript.

The second paragraph in section: **Experiments on Zinc**. It reads:

~~t-SMILES based models significantly outperforms another fragment dictionary based model, FragDgm, which splits molecule in a linear mode as a sequence of fragment IDs, on all five~~

~~distribution parameters. Although FragDgm uses a segmented mode and being based on distributional learning, its FCD value of 0.303 is the lowest among all listed models.~~

2. CReM, FASMIFRA (the second reference) and eSynth (a related algorithm in the first reference) use different ways to assemble fragments, they all belong to a big category of fragment assembling algorithms that could serve as a part of the t-SMILES framework as a reconstruction algorithm.

This kind solutions seem to struggle with ‘the limitation of **inability to create new ring systems** so the performance depends on their representativeness in the input compound database.’ On the contrary, t-SMILES model **can effectively generate new rings and new fragments**. From this point of view, they are rather a complementary one with t-SMILES.

Due to the total volume of this study, this broad category of algorithms was not originally extensively covered in the scope of this study. Currently, although only random algorithms are still used to select candidates, even in goal-directed reconstruction methods, t-SMILES models outperform baseline models by a wide margin in goal-directed tasks. In future research, it may be worthwhile to consider using any of them to reconstruct fragments in the t-SMILES process for improved performance. As you suggested, further exploration could be considered in a future paper once t-SMILES have been established.

We thank the reviewer for their helpful feedback, constructive criticism and their suggestion to include more references, which has helped us to improve the quality of our manuscript and provide a more comprehensive comparison of our solution with other state-of-the-art models.

Reviewer #2 (Remarks to the Author):

I appreciate the authors' considered responses to my comments. All my comments have been addressed.

Response 2.0

We would like to express our deep appreciation to the reviewers for their review of our manuscript.

Reference

- [1] F. Berenger and K. Tsuda, "Molecular generation by Fast Assembly of (Deep)SMILES fragments," *J. Cheminform.*, vol. 13, no. 1, pp. 1–10, Dec. 2021, doi: 10.1186/S13321-021-00566-4/FIGURES/6.
- [2] M. Naderi, C. Alvin, Y. Ding, S. Mukhopadhyay, and M. Brylinski, "A graph-based approach to construct target-focused libraries for virtual screening," *J. Cheminform.*, vol. 8, no. 1, pp. 1–16, Mar. 2016, doi: 10.1186/S13321-016-0126-6/FIGURES/11.
- [3] S. Jinsong, J. Qifeng, C. Xing, Y. Hao, and L. Wang, "Molecular fragmentation as a crucial step in the AI-based drug development pathway," *Commun. Chem.*, vol. 7, no. 1, p. 20, Feb. 2024, doi: 10.1038/s42004-024-01109-2.
- [4] T. Liu, M. Naderi, C. Alvin, S. Mukhopadhyay, and M. Brylinski, "Break Down in Order to Build Up: Decomposing Small Molecules for Fragment-Based Drug Design with eMolFrag," *J. Chem. Inf. Model.*, vol. 57, no. 4, pp. 627–631, 2017, doi: 10.1021/acs.jcim.6b00596.

REVIEWERS' COMMENTS

Reviewer #1 (Remarks to the Author):

Publish as is.